# Effect of Saturation on Shear Behavior and Particle Breakage of Coral Sand

Xiang Chen [1], Jianhua Shen [1,*], Xing Wang [2,3], Ting Yao [1] and Dongsheng Xu [4]

1 State Key Laboratory of Geotechnical Mechanics and Engineering, Institute of Rock and Soil Mechanics, Chinese Academy of Sciences, Wuhan 430071, China
2 School of Civil Engineering, Guangzhou University, Guangzhou 510006, China
3 Earthquake Engineering Research & Test Center, Guangzhou University, Guangzhou 510405, China
4 School of Civil Engineering and Architecture, Wuhan University of Technology, Wuhan 430062, China
* Correspondence: jhshen@whrsm.ac.cn

**Abstract:** Coral sand is the main filling material for the island–reef foundation. Under tidal actions, the saturation ($Sr$) of coral sand layers varies with the specific depths in the reclaimed foundation. Studying the $Sr$ effect of coral sand's mechanical behaviors is crucial for the stability of the reclaimed foundation of island–reefs. In this study, a "quantitative injection method" was designed to prepare coral sand with saturation ranging from 90% to 100%, and unconsolidated–undrained (UU) triaxial shear tests were conducted on coral sand under different effective confining pressures ($\sigma_3'$). The results indicated that the stress–strain curves of coral sand under various conditions were of the strain-softening type. When $\sigma_3'$ = 200, 400, 600, and 800 kPa, the shear strength of coral sand decreased exponentially by 13.1, 9.1, 16.8, and 15.2%, respectively, with the increase in $Sr$ from 90% to 100%. As $Sr$ rose, the internal friction angle ($\varphi$) dropped by 3.77°. The cohesion ($c$) was not significantly affected by $Sr$ compared to $\varphi$. In consideration of the physical susceptibility of coral sand to breakage, relative breakage ratio ($B_r$) and modified relative breakage index ($B_r^*$) were introduced to evaluate the particle breakage behaviors of coral sand samples with different $Sr$ levels in the triaxial shear process. It was found that $B_r$ and $B_r^*$ increase linearly with increasing $Sr$; the effect of $Sr$ on the particle breakage of coral sand weakens significantly when $\sigma_3'$ is sufficiently large. The median particle size ($d_{50}$) of coral sand decreases with increasing $Sr$, and presents a negative linear correlation with both $B_r$ and $B_r^*$. Based on comparing the strength and particle breakage characteristics of coral sand samples with varying $Sr$ levels, this study suggests that 92.5% should be considered as the $Sr$ value of coral sand available for testing.

**Keywords:** coral sand; saturation; effective confining pressure; shear behavior; particle breakage

## 1. Introduction

Coral sand, also known as calcareous sand, is a geomaterial composed primarily of $CaCO_3$ [1,2]. Coral sand is mainly distributed on both sides of the equator in the Pacific Ocean, the Indian Ocean, the Atlantic Ocean, and the South China Sea [3]. In recent years, China has launched large-scale island–reef reclamation construction projects in the South China Sea [4]. According to its natural geographical advantages (i.e., island–reefs in the South China Sea are far from the mainland) and excellent engineering characteristics, coral sand has become the preferred material for island–reef foundation filling and infrastructure construction [5–7]. The reclaimed layers of island–reefs are only 3–5 m thick [8]. Under tidal actions, coral sand, as the filling material of bearing strata, is generally in a highly saturated (but not fully saturated) state [9]. Therefore, studying the effects of saturation ($Sr$) on the engineering mechanical properties of coral sand is crucial for the stability of the reclaimed foundation of island–reefs.

Triaxial shear tests can simulate the stress state of geomaterials in strata [10]. Therefore, predecessors mostly used triaxial shear tests to study the engineering mechanical properties

of coral sand and achieved some research results (Table 1) [11–18]. However, predecessors primarily utilized saturated coral sand as the research object but rarely touched upon the engineering mechanical properties of unsaturated coral sand [19]. Coral sand has a poor water-holding capacity [20], and it is difficult to precisely arrange coral sand samples with different $Sr$ levels [21,22]. Water loss is also inevitable when filling coral sand with design $Sr$ into the mold. These reasons may explain why predecessors have infrequently explored the engineering mechanical properties of unsaturated coral sand. Before investigating the engineering mechanical properties of coral sand samples with different $Sr$ levels based on triaxial shear tests, the first critical technical problem to be solved is to develop a new sample preparation method. Next, if coral sand undergoes isotropic compression prior to triaxial shear tests, pore water will inevitably overflow from the sample, decreasing $Sr$. It will also be impossible to monitor the change in the $Sr$ of coral sand during this process using conventional triaxial shear test apparatus. Therefore, it is preferable to investigate the engineering mechanical properties of coral sand samples with varying $Sr$ using unconsolidated–undrained (UU) triaxial shear tests.

**Table 1.** Previous research on mechanical properties of coral sand based on triaxial shear test.

| Research Material | Research Content | Research Method | Researcher and Year | Shear Type | Research Variable | Main Conclusions |
|---|---|---|---|---|---|---|
| Saturated coral sand | Mechanical properties | Triaxial shear test | Sharma and Ismail (2006) [11] | CU | Soil origin, Relative density, Initial mean effective stress | The monotonic shear response of calcareous soil from Goodwyn and Ledge Point was similar to that of siliceous sand, and the peak friction angel were relatively higher than siliceous sand. |
| | | | Hassanlourad et al. (2014) [12] | CU | Effective confining pressure, Relative density | Coral sand illustrated more shear strength than quartz sand in the triaxial consolidated undrained test. |
| | | | Zhang et al. (2019) [13] | CD | Fine particle content | The dilatancy property and peak deviator stress of coral sand decreased with the increasing fine particle content. |
| | | | Wu et al. (2020) [14] | CD | Effective confining pressure, Relative density | The strain softening and dilatation characteristic of coral sand gradually weaken with increasing effective confining pressure and decreasing compactness. |
| | | | Saeidaskari et al. (2021) [15] | CU | Effective confining pressure, Relative density | Coral sand showed initial contractive behavior, and the contractive behavior became more apparent as the effective confining pressure increased. |
| | | | Liu et al. (2022) [16] | CU | Effective confining pressure, Relative density | The shear modulus of coral sand increased with effective confining pressures and relative density, and the shear strength (or internal frictional angle) in peak state was more than that in critical state. |
| | | | Wang et al. (2022) [17] | CD | Particle size, Effective confining pressure | The shear strength of coral sand specimens with a single particle size decreased with an increase in particle size. |
| | | | Chen et al. (2022) [18] | CD | Particle gradation, Effective confining pressure | The softening and dilatancy were more significant for coral sand with smaller particle size, and the average peak friction angle of coral sand decreased linearly with the increasing mean particle size. |

Moreover, coral sand is a special geomaterial susceptible to particle breakage even under low stress [23]. Predecessors also studied the particle breakage characteristics of coral sand, relying on triaxial shear tests (Table 2) [14,18,24–29]. They discovered how factors such as particle size, stress, and compactness affect the particle breakage of coral sand. However, the impact of *Sr* on the particle breakage of coral sand is seldom reported [30,31]. Indeed, the stress state significantly affects the particle breakage of coral sand [32,33]. It should be noted that the above findings are primarily obtained from consolidated-drained (CD) or consolidated-undrained (CU) triaxial shear tests and that the stress state of coral sand in UU triaxial shear tests differs greatly from CD or CU. Accordingly, the applicability of the above results to coral sand in UU triaxial shear tests must still be confirmed by solid test data, and related research should be expanded.

**Table 2.** Pervious research on particle breakage of coral sand based on triaxial shear test.

| Research Material | Research Content | Research Method | Researcher and Year | Shear Type | Research Variable | Main Conclusions |
|---|---|---|---|---|---|---|
| Saturated coral sand | Particle breakage | Triaxial shear test | Zhang et al. (2008) [24] | CD | Effective confining pressure, Axial strain | The increase of effective confining pressure and axial strain could promote the particle breakage of coral sand. |
| | | | Shahnazari and Rezvani (2013) [25] | CD, CU | Effective confining pressure, Relative density, Axial strain, Drainage condition, Grain size distribution | Increasing of effective confining pressure, axial strain, relative density and grain size resulted in a higher particle breakage for coral sand, and coral sand under undrained condition had less particle breakage compared to the drained experiments due to the increasing of pore pressure. |
| | | | Yu (2018) [26] | CD, CU | Effective confining pressure, Initial void ratio, Consolidated stress ratio | In isotropic consolidation, more particle breakage of coral sand was cause in higher effective confining pressure (or denser sample), and anisotropic stress yielded more particle breakage than isotropic stress for coral sand. |
| | | | Wu et al. (2020) [14] | CD | Effective confining pressure, Relative density | The relationship between particle breakage and plastic work of coral sand could be described by a power function with a negative index, and particle breakage increased with the increasing plastic work with a hyperbolic form. |
| | | | Wang et al.(2021) [27] | CD, CU | Effective confining pressure, Axial strain | Particle breakage of coral sand increased with axial strain at a gradually decreasing rate during triaxial drained compression, and shear strain could induce further breakage without an increase in stress. |
| | | | Wang et al.(2021) [28] | CD | Effective confining pressure, Relative density | For coral gravelly sand, the highest breakage degree was observed in sand particles within 1–2 mm size range. |
| | | | Chen et al. (2022) [18] | CD | Particles gradation Effective confining pressure | The particle breakage of coral sand increased with the increasing effective confining pressure and mean particle size. |
| | | | Shen et al. (2022) [29] | CU | Effective confining pressure | The particle breakage of coral sand increased in powder function form with increasing effective confining pressure, and splitting and abrasion were the main particle breakage patterns of coral sand within 400 kPa effective confining pressure. |

In light of this, an independently designed "quantitative injection method" was used in this study to prepare coral sand samples with various *Sr* levels. The UU triaxial shear tests were conducted on these samples under different effective confining pressures ($\sigma_3'$) to ascertain the *Sr* effect of the coral sand shear behavior. Finally, screening tests were performed on tested samples to understand the particle breakage behaviors of coral sand samples with different *Sr* levels during UU triaxial shear tests. The study's results provide theoretical support for assessing the stability of the reclaimed foundation of island–reefs.

## 2. Test Overview

### 2.1. Material

This study uses uncemented loose coral sand collected from a hydraulically reclaimed island–reef in the South China Sea. Figure 1 exhibits the uniform initial gradation of coral sand with particle sizes ranging from 0.25 to 0.5 mm. Its nonuniformity coefficient ($Cu$) and curvature coefficient ($Cc$) are 1.455 and 0.96, respectively (Table 3). According to the *Chinese National Standard of Soil Test Method* (GB/T 50123–2019), it should be named as poorly graded coral medium sand. The specific gravity ($Gs$), maximum dry density ($\rho_{dmax}$), and minimum dry density ($\rho_{dmin}$) of coral sand are 2.83, 1.385, and 1.085, respectively. According to GB/T 50123–2019, the minimum void ratio ($e_{min}$) and maximum void ratio ($e_{max}$) of calcareous sand are 1.043 and 1.608, respectively.

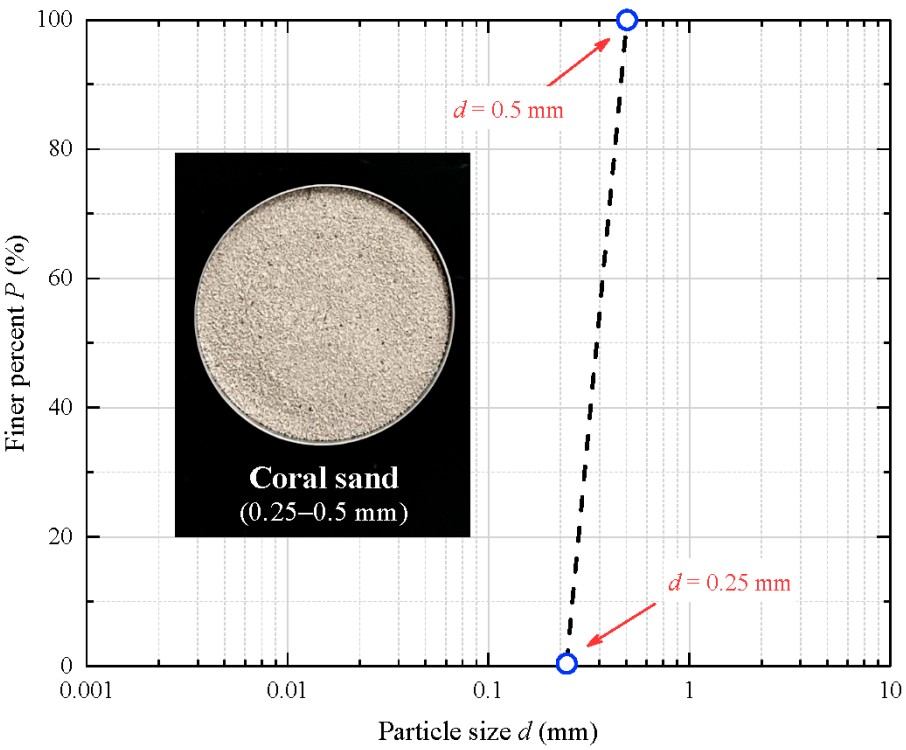

**Figure 1.** Particle distribution curve of coral sand.

**Table 3.** Physical parameters of coral sand.

| Sample | Particle Size $d$ (mm) | Median Size $d_{50}$ (mm) | Nonuniform Coefficient $Cu$ | Curvature Coefficient $Cc$ | Specific Gravity $Gs$ | Maximum Void Ratio $e_{max}$ | Minimum Void Ratio $e_{min}$ | Maximum Dry Density $\rho_{dmax}$ (g/cm³) | Minimum Dry Density $\rho_{dmin}$ (g/cm³) |
|---|---|---|---|---|---|---|---|---|---|
| Coral medium sand | 0.25–0.5 | 0.375 | 1.455 | 0.96 | 2.83 | 1.608 | 1.043 | 1.385 | 1.085 |

### 2.2. Test Scheme

This study analyzes the effects of $Sr$ on the shear behavior and particle breakage behavior of coral sand using UU triaxial shear tests. The selection of $Sr$ was determined by considering the following two assumptions: (I) All the pores in coral sand are in a closed state, and the effect of the inner pores released due to particle breakage in the shear process on test results can be ignored; (II) the $Sr$ of coral sand is 100% when interparticle pores are filled up with water. Considering these assumptions and the nearly saturated state of coral sand in most reclaimed layers, the $Sr$ of coral sand was set at 100, 97.5, 95, 92.5,

and 90% utilizing a gradient of 2.5%, respectively. The *Sr* effect of the shear behavior of coral sand was studied using UU triaxial shear tests to avoid the change in the *Sr* of coral sand caused by water loss during consolidation and shear processes. After foundation treatment, coral sand can generally reach a compact state [28]. Hence, coral sand in shear tests was set with a relative density (*Dr*) of 70% to simulate its actual physical state in strata as realistically as possible. The $\sigma_3'$ applied to coral sand was 200, 400, 600, and 800 kPa. These $\sigma_3'$ settings covered multiple environments from low to high stress and were intended to investigate the *Sr* effect of the shear behavior of coral sand over a broad stress range. All coral sand samples were Ø39.1 × 80 mm in size (diameter × height). Shearing proceeded at a 0.15 mm/min rate and stopped when axial strain ($\varepsilon_a$) reached 30% (Table 4). It is noteworthy that coral sand is a geomaterial highly susceptible to breakage [34]. Screening tests were performed on samples to assess the particle breakage behaviors of coral sand samples with different *Sr* levels during the UU triaxial shear process.

**Table 4.** Triaxial shear test scheme.

| Sample | Sample No. | Saturation $Sr$ (%) | Relative Density $Dr$ (%) | Specimen Size | | Effective Confining Pressure $\sigma_3'$ (kPa) | Shear Type | Shear Rate $v$ (mm/min) | Terminal Strain $\varepsilon_t$ (%) |
| | | | | Diameter (mm) | Height (mm) | | | | |
|---|---|---|---|---|---|---|---|---|---|
| Coral medium sand | CS–90 | 90 | 70 | 39.1 | 80 | 200 | UU | 0.15 | 30 |
| | CS–90 | 90 | 70 | 39.1 | 80 | 400 | UU | 0.15 | 30 |
| | CS–90 | 90 | 70 | 39.1 | 80 | 600 | UU | 0.15 | 30 |
| | CS–90 | 90 | 70 | 39.1 | 80 | 800 | UU | 0.15 | 30 |
| | CS–92.5 | 92.5 | 70 | 39.1 | 80 | 200 | UU | 0.15 | 30 |
| | CS–92.5 | 92.5 | 70 | 39.1 | 80 | 400 | UU | 0.15 | 30 |
| | CS–92.5 | 92.5 | 70 | 39.1 | 80 | 600 | UU | 0.15 | 30 |
| | CS–92.5 | 92.5 | 70 | 39.1 | 80 | 800 | UU | 0.15 | 30 |
| | CS–95 | 95 | 70 | 39.1 | 80 | 200 | UU | 0.15 | 30 |
| | CS–95 | 95 | 70 | 39.1 | 80 | 400 | UU | 0.15 | 30 |
| | CS–95 | 95 | 70 | 39.1 | 80 | 600 | UU | 0.15 | 30 |
| | CS–95 | 95 | 70 | 39.1 | 80 | 800 | UU | 0.15 | 30 |
| | CS–97.5 | 97.5 | 70 | 39.1 | 80 | 200 | UU | 0.15 | 30 |
| | CS–97.5 | 97.5 | 70 | 39.1 | 80 | 400 | UU | 0.15 | 30 |
| | CS–97.5 | 97.5 | 70 | 39.1 | 80 | 600 | UU | 0.15 | 30 |
| | CS–97.5 | 97.5 | 70 | 39.1 | 80 | 800 | UU | 0.15 | 30 |
| | CS–100 | 100 | 70 | 39.1 | 80 | 200 | UU | 0.15 | 30 |
| | CS–100 | 100 | 70 | 39.1 | 80 | 400 | UU | 0.15 | 30 |
| | CS–100 | 100 | 70 | 39.1 | 80 | 600 | UU | 0.15 | 30 |
| | CS–100 | 100 | 70 | 39.1 | 80 | 800 | UU | 0.15 | 30 |

### 2.3. Apparatus and Procedure

The test apparatus is a TKA–Advanced full-automatic stress path triaxial testing apparatus manufactured by Nanjing TKA Technology Co., Ltd. (Nanjing, China) (Figure 2). It mainly consists of a pressure chamber, a displacement sensor, a load sensor, a data acquisition system, a strain control system, and pressure/volume controllers (used to control confining and back pressure). The apparatus supports variable-speed loading and has a shear rate range of 0.0001–9.9999 mm/min. The load sensor has a range of 10 kN and a precision of 0.15% F.S. The two pressure/volume controllers have a stress control range of 0–2000 kPa (precision = 1 kPa), and a volume range of 250 mL (precision = 0.001 mL). The LCD screen on each pressure/volume controller displays the current confining pressure/back pressure and the volume of water inside. The on-screen keyboard can be employed to regulate stress and absorb or drain water by the specified volume.

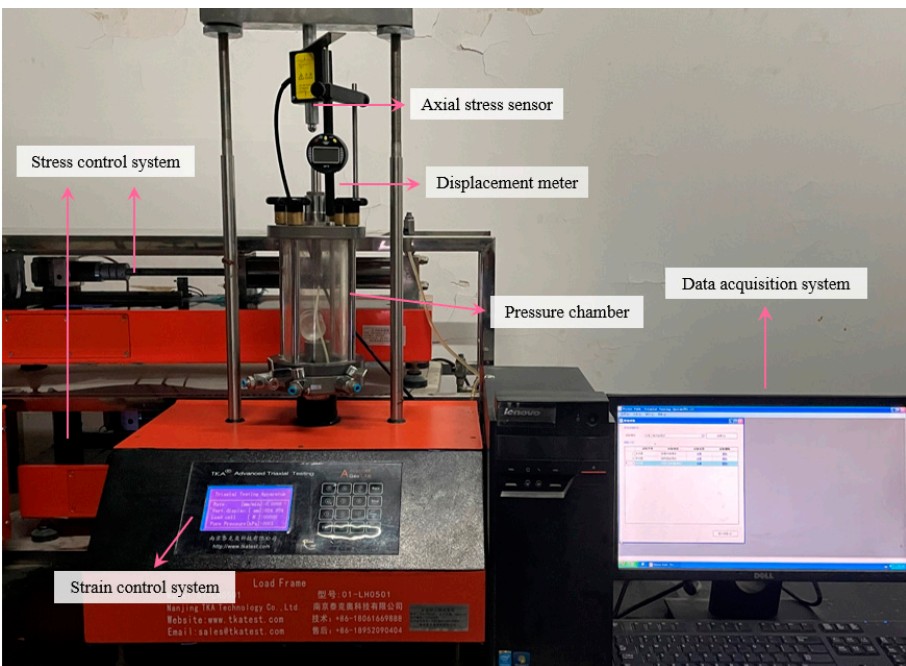

**Figure 2.** Full-automatic stress path triaxial apparatus.

Precise control of *Sr* is the most critical step in studying the *Sr* effect on the shear behavior of coral sand. Most predecessors control the water content of a sample by first preparing a sample with designed water content and then transferring the sample into a rubber membrane [35–37]. This approach is ideal for regulating the water content of fine-grained soils (such as clay and silt). Due to the poor water-holding capacity of coral sand, it is difficult to prevent water overflow when loading a sample of coral sand with designed water content into the rubber membrane. Therefore, an original method was designed to prepare coral sand samples with different *Sr* levels as follows.

(1) The dry density of the sample under 70% *Dr* was determined utilizing the physical parameters listed in Table 1. The mass of the dry sample was calculated from dry density and sample size, and dry coral sand was weighed accordingly (Figure 3a). An impermeable stone was placed on the base of the pressure chamber, sheathed by the rubber membrane, and fixed using a split mold. A vacuum pump was connected to the interface outside of the split mold to remove the gas between the rubber membrane and the split mold's inner wall, allowing the rubber membrane to perfectly fit with the split mold's inner wall (Figure 3b).

(2) The sample was loaded into the rubber membrane by five layers through sand alluviation. Each time a layer was loaded, a graduated scale was used to measure the distance between the sample surface and the split mold top to ensure the compactness of the sample matched that in Table 2 (Figure 3c). The sample surface was roughened before the next layer was added. The above steps were repeated until the sample's upper surface reached the same level as the top of the split mold. The sample cap was covered after the completion of sample loading. At this point, the preparation of a dry coral sand sample with 70% *Dr* was completed.

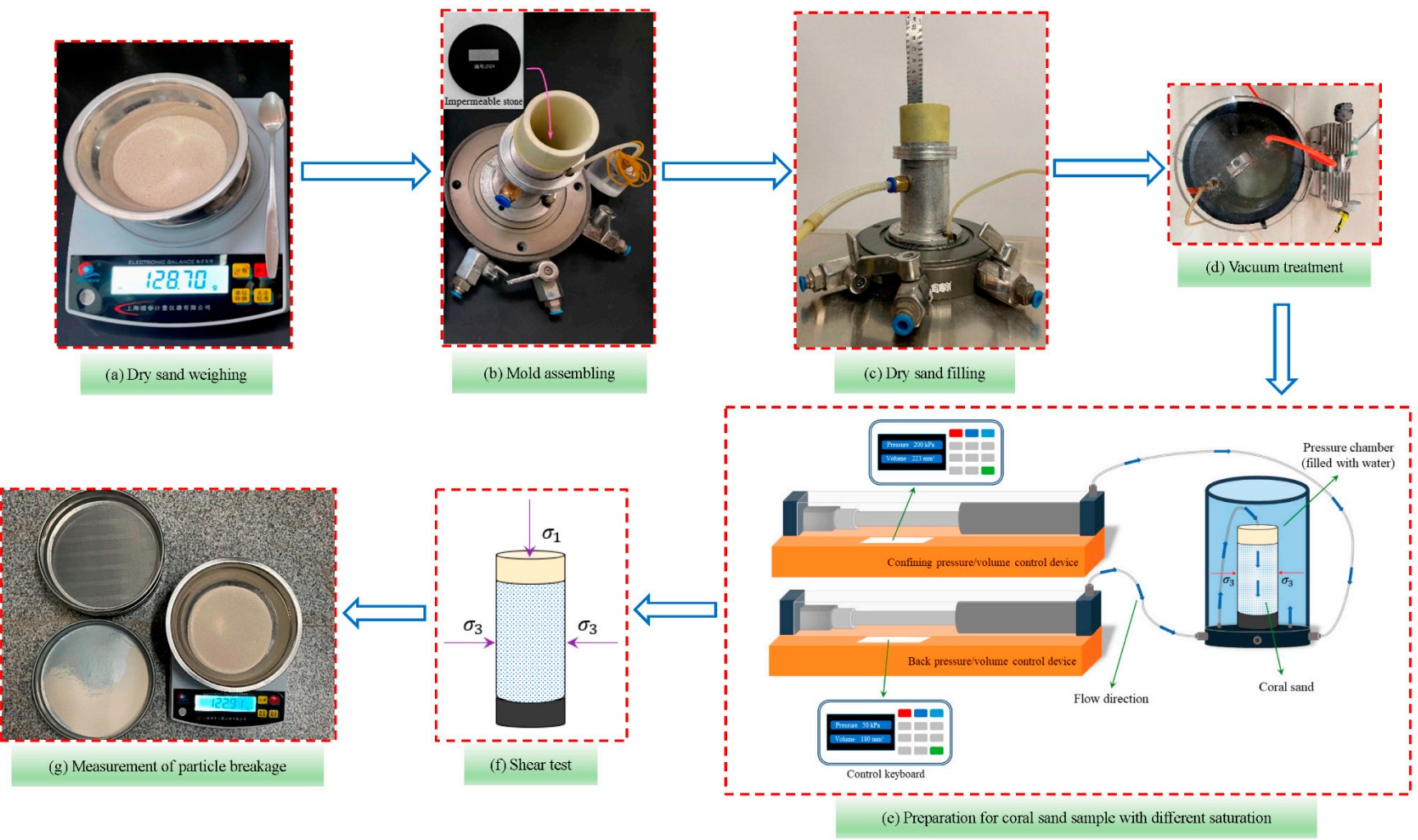

**Figure 3.** Diagrammatic sketch of test procedures.

(3) The theoretical water contents of coral sand samples with different *Sr* levels were calculated from the physical parameters and test parameters indicated in Tables 1 and 2 as follows:

$$Dr = \frac{e_{max} - e}{e_{max} - e_{\min}} \tag{1}$$

$$n = \frac{e}{e + 1} \tag{2}$$

where *n* is porosity. The void ratio *e* of the coral sand sample with 70% *Dr* was calculated by Equation (1) and then substituted into Equation (2) to determine porosity *n*. The volume of water in the sample was computed as follows:

$$V_W = S_r \cdot V_V = S_r(n \cdot V) \tag{3}$$

where $V_W$ is the volume of water in the sample, $V_V$ is the volume of pores in the sample, and *V* is the total volume of the sample. The water contents of coral sand samples with different *Sr* levels were calculated by Equation (3) (Table 5). First, the sample was vacuum-treated to remove the gas inside (Figure 3d) and produce a negative pressure (to increase the seepage rate of water in the sample in the subsequent water injection process). Next, the pressure chamber was placed on the base of the triaxial shear apparatus, and water was injected into the pressure chamber so that the sample cap would come into contact with the load sensor. The confining pressure in the pressure chamber was set at 200 kPa. Multiple trial tests confirmed that a confining pressure of 200 kPa can prevent the volume expansion caused by increasing back pressure during the water injection process. After the confining pressure stabilized, the back pressure valve at the base of the pressure chamber was opened. The back pressure/volume controller was utilized to inject water into the sample until the corresponding *Sr* was achieved (Figure 3e). Due to the gradual filling of pores by water, back pressure will develop in the sample at this time. Finally, after the water injection was completed, the sample was left to rest for two hours, after which the back pressure value displayed on the back pressure/volume controller was recorded (denoted as *x* kPa). This value represents the back pressure value prior to shearing. The effective confining pressure was denoted as *y* kPa, in which case the actual confining pressure will be (*x* + *y*) kPa.

**Table 5.** Water content of coral sand sample with different saturations.

| Sample | Sample No. | *Sr* (%) | Water Content (mL) |
|---|---|---|---|
| | CS–90 | 90 | 45.56 |
| | CS–92.5 | 92.5 | 46.82 |
| Coral medium sand | CS–95 | 95 | 48.09 |
| | CS–97.5 | 97.5 | 49.35 |
| | CS–100 | 100 | 50.62 |

(4) UU triaxial shear tests were conducted on coral sand samples under design $\sigma_3'$. Shearing was performed at a 0.15 mm/min rate and stopped when $\varepsilon_a$ = 30% (Figure 3f).

(5) After shear tests, samples were washed with clean water into the tray, which was placed in an oven with a constant temperature of 105 °C. It was removed after 24 h and allowed to cool down to room temperature. Then, screening tests were conducted on the samples to examine their particle gradation curves (Figure 3g). An excessively short screening time causes inadequate screening while an excessively long one aggravates particle breakage. As a result, the screening time was set at 15 min each time [24].

## 3. Shear Test Result and Analysis

### 3.1. Deviator Stress–Pore Water Stress–Axial Strain Relationship

Figure 4 shows the deviator stress–axial strain curves (i.e., $q$–$\varepsilon_a$ curves) of coral sand samples under different conditions. It can be seen from Figure 4 that, within the $\sigma_3'$ range of

800 kPa, the $q$–$\varepsilon_a$ curves of coral sand samples with 90–100% $Sr$ during the UU triaxial shear stress state presented a strain softening trend. The $q$–$\varepsilon_a$ curve of each coral sand sample can be divided into three stages:

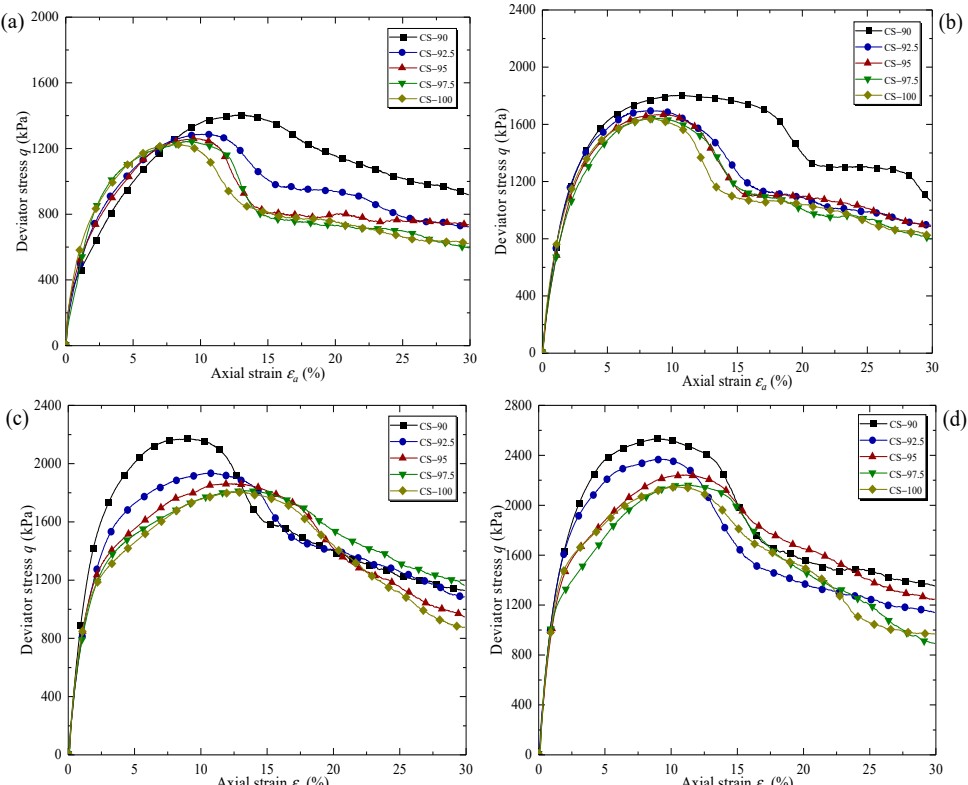

**Figure 4.** Deviator stress-axial strain curves of coral sand with different saturations: (**a**) $\sigma_3' = 200$ kPa; (**b**) $\sigma_3' = 400$ kPa; (**c**) $\sigma_3' = 600$ kPa; (**d**) $\sigma_3' = 800$ kPa.

(1) Elastic stage: In the initial shear stage, $q$ surged with increasing $\varepsilon_a$, and the $q$–$\varepsilon_a$ curve was almost linear.

(2) Plastic yield stage: During the intermediate shear stage, $q$ continued to grow with increasing $\varepsilon_a$, but at a gradually slower rate, and the $q$–$\varepsilon_a$ curve presented a gentle changing trend.

(3) Failure stage: In the late shear stage, due to increasing $\varepsilon_a$, $q$ first peaked and then continued to decrease until the test was complete. The sample showed evident strain softening characteristics.

Figure 5 depicts the pore water stress–axial strain curves (i.e., $u$–$\varepsilon_a$ curves) of coral sand samples under different conditions. When $\sigma_3' = 200$ kPa, $u$ was negative, and the sample exhibited a dilatancy similar to dense sand [38]. When $\sigma_3' = 400$ kPa, only CS–100 showed obvious dilation. When $\sigma_3' = 600$ and 800 kPa, $u$ was not negative, and the sample was always in a state of contraction. First, increasing $\sigma_3'$ tightened the constraint on the sample and prevented its dilation deformation [39]. Second, increasing $\sigma_3'$ aggravated the particle breakage of the sample in the shear process and reduced its dilation degree [40]. Therefore, when $\sigma_3' = 600$ and 800 kPa, the sample remained in a state of contraction throughout the shear process.

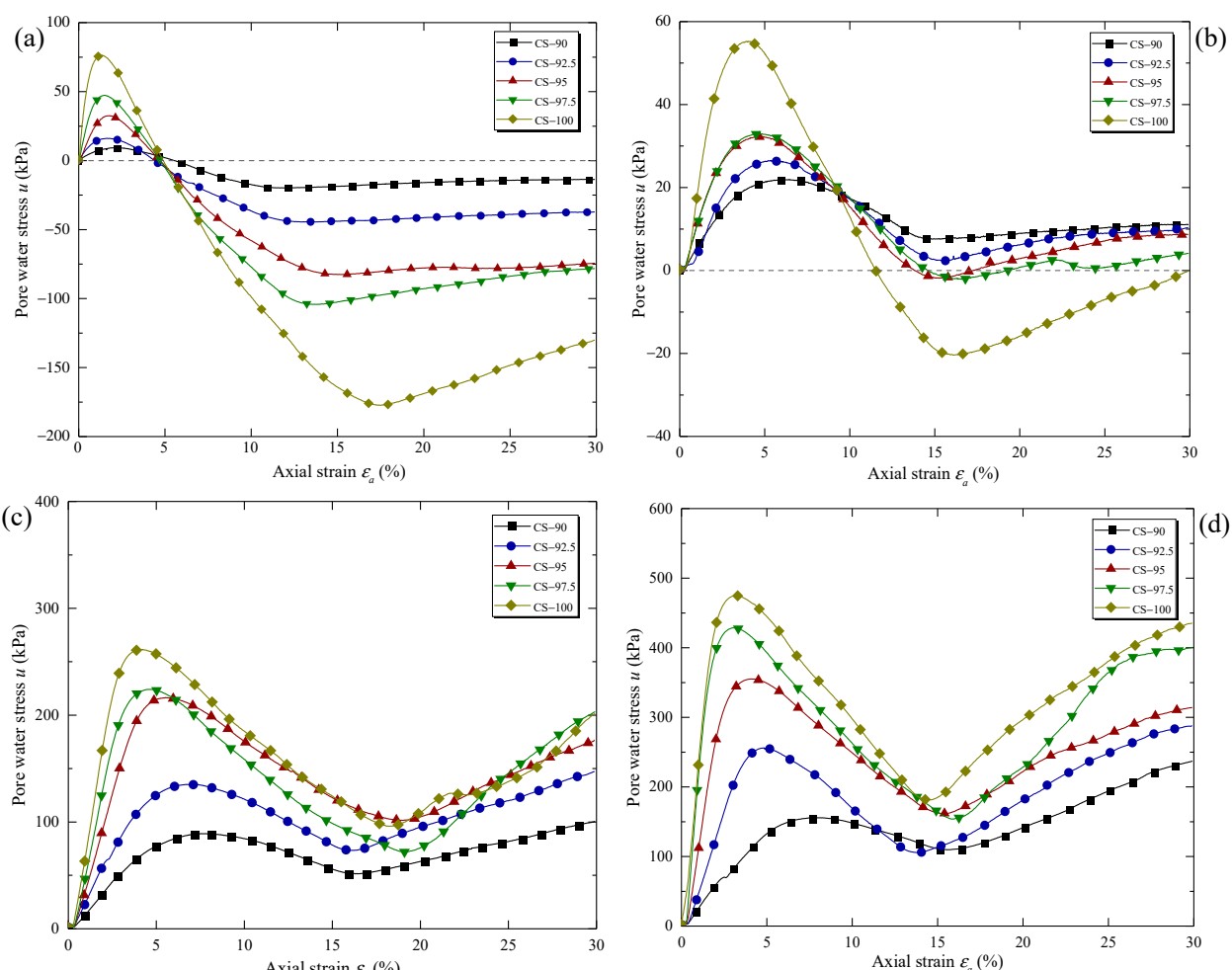

**Figure 5.** Pore water stress-axial strain curves of coral sand with different saturations: (**a**) $\sigma'_3$ = 200 kPa; (**b**) $\sigma'_3$ = 400 kPa; (**c**) $\sigma'_3$ = 600 kPa; (**d**) $\sigma'_3$ = 800 kPa.

Within the $\sigma'_3$ range of 800 kPa, $u-\varepsilon_a$ curves of coral sand samples with 90–100% $Sr$ during the UU triaxial shear stress state can also be separated into three stages:

(1) In the initial shear stage, $u$ rose rapidly with increasing $\varepsilon_a$, and the sample contracted. In the initial UU triaxial shear stage, the sample's pore water produced $u$ under external loads. Growing $\varepsilon_a$ (or shear stress $\tau$) increased the mean effective stress ($p'$) borne by the sample, resulting in sample contraction. Sample contraction also enhanced the squeezing action on pore water and caused $u$ to present a growing trend with rising $\varepsilon_a$ [41]. The peak pore water stress ($u_{p1}$) can be utilized to characterize the maximum contraction degree of coral sand samples. The curve of $u_{p1}$ changing with $Sr$ was plotted to explore the effect of $Sr$ on $u$ of a coral sand sample in the initial shear stage (Figure 6). According to Figure 6: (I) there is an apparent positive correlation between $u_{p1}$ and $Sr$ under the same $\sigma'_3$. A sample with a higher $Sr$ level contains more pore water content and also produces a more significant $u$ under the constraint of $\sigma'_3$. A positive correlation between $u_{p1}$ and $Sr$ indicates that a sample with a higher $Sr$ level has a greater contraction degree in the initial shear stage. (II) For coral sand samples with the same $Sr$, $u_{p1}$ increases more significantly under high effective confining pressures ($\sigma'_3$ = 600 and 800 kPa) than under low effective confining pressures ($\sigma'_3 \leq 400$ kPa), and the increments also increase with increasing $Sr$. This phenomenon was mainly caused by the constraint imposed by $\sigma'_3$ on coral sand samples. Under a low effective confining pressure ($\sigma'_3 \leq 400$ kPa), the constraining force was weak, and $u$ was low and insensitive to the change in $Sr$. Increasing $\sigma'_3$ enhanced the constraining force, caused the increase of $u$, and amplified the effect of $Sr$ on $u$.

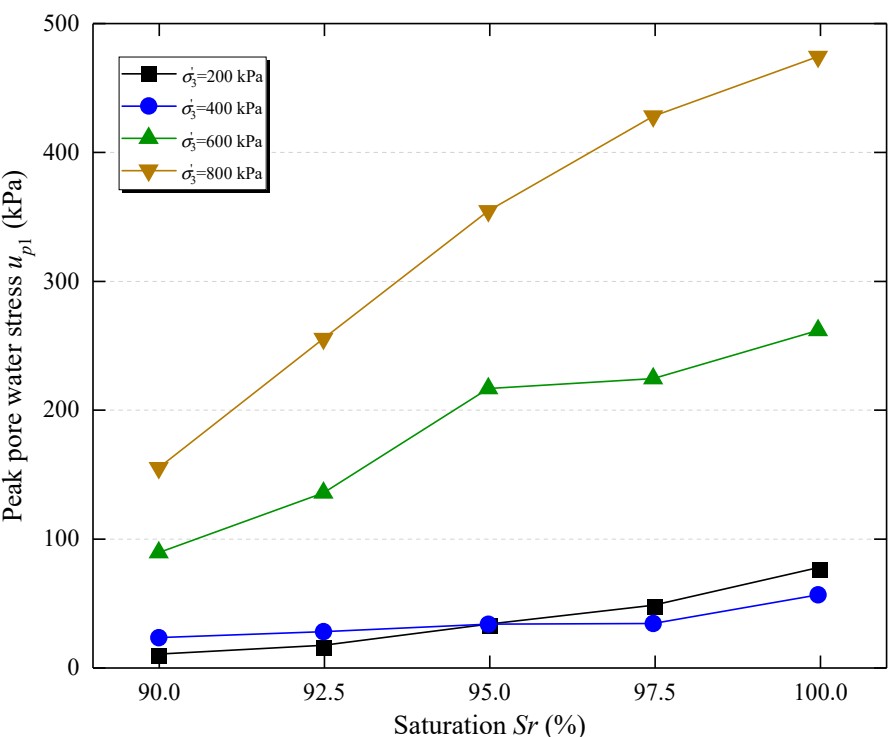

**Figure 6.** Relationship between peak pore water stress and saturation of coral sand.

(2) $u$ gradually decreased with increasing $\varepsilon_a$ after reaching $u_{p1}$. This was primarily due to the expansion of coral sand particles in the shear process. During the dilation process, coral sand particles bore more external loads. Moreover, the external loads carried by coral sand particles increased with increasing dilation degree, resulting in a decreasing trend for $u$.

(3) $u$ increased with increasing $\varepsilon_a$ after reaching the minimum (at which point coral sand particles had the largest dilation degree). The failure modes of coral sand samples with various $Sr$ levels were essentially identical. Due to space limitations, Figure 7 only shows the post-test forms of coral sand samples with different $Sr$ levels when $\sigma_3' = 200$ kPa. In the late shear stage, a shear band emerged gradually in coral sand samples (that is, local shear occurred) and slowly expanded with increasing $\varepsilon_a$. This was the fundamental reason why $u$ increased with increasing $\varepsilon_a$ after reaching the minimum.

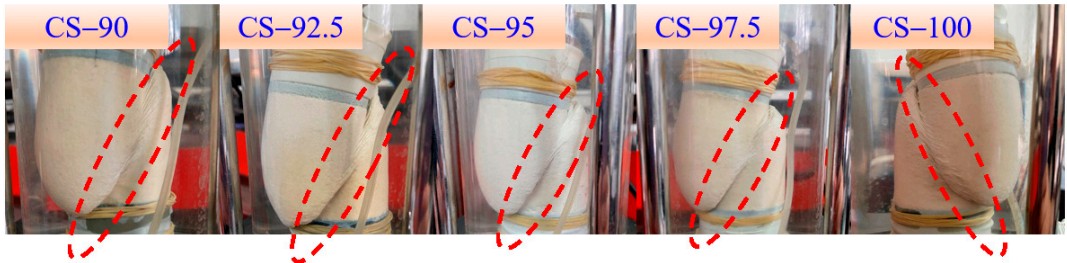

**Figure 7.** Failure pattern of coral sand samples after shear test.

*3.2. Stress Path*

Stress path curves are curves that describe the process of $q'$ changing with $p'$ in the triaxial shear process. In UU triaxial shear stress state, the $q'$ and $p'$ of a geomaterial can be expressed as:

$$q' = \sigma_1' - \sigma_3' \tag{4}$$

$$p' = \frac{\sigma_1' + 2\sigma_3'}{3} \tag{5}$$

where $\sigma_1'$ and $\sigma_3'$ are effective major and minor principal stresses, respectively. When $u$ reached the minimum, the sample experienced local shear (or the development of sample deformation was no longer homogeneous), so this study only discussed the stress paths of coral sand samples before $u$ reached the minimum (Figure 8). As seen in Figure 8, coral sand samples with different $Sr$ levels have similar stress paths under a low effective confining pressure ($\sigma_3' \leq 400$ kPa). In the UU triaxial shear process, the difference in $u$ was the macroscopic mechanism for which coral sand samples with various $Sr$ levels showed different stress responses under the same $\sigma_3'$. However, the change in $Sr$ did not significantly affect the development of stress in coral sand samples, mainly because $u$ was significantly lower than principal stresses (or there was no noticeable difference in $u$ between coral sand samples with different $Sr$ levels) under a low $\sigma_3'$. For example, when $\sigma_3' = 200$ kPa, the maximum deviation in $u_{p1}$ between coral sand samples with different $Sr$ levels was only 67.2 kPa. For this reason, the stress path curves of coral sand samples with different $Sr$ levels showed no discernible differences from each other. When $\sigma_3'$ increased to 600 kPa (or 800 kPa), the maximum deviation in $u_{p1}$ between coral sand samples with different $Sr$ levels reached 173 kPa (or 320.1 kPa). As a result, the stress path curves of coral sand samples with many $Sr$ levels differed significantly from each other under high effective confining pressures ($\sigma_3' = 600$ and 800 kPa).

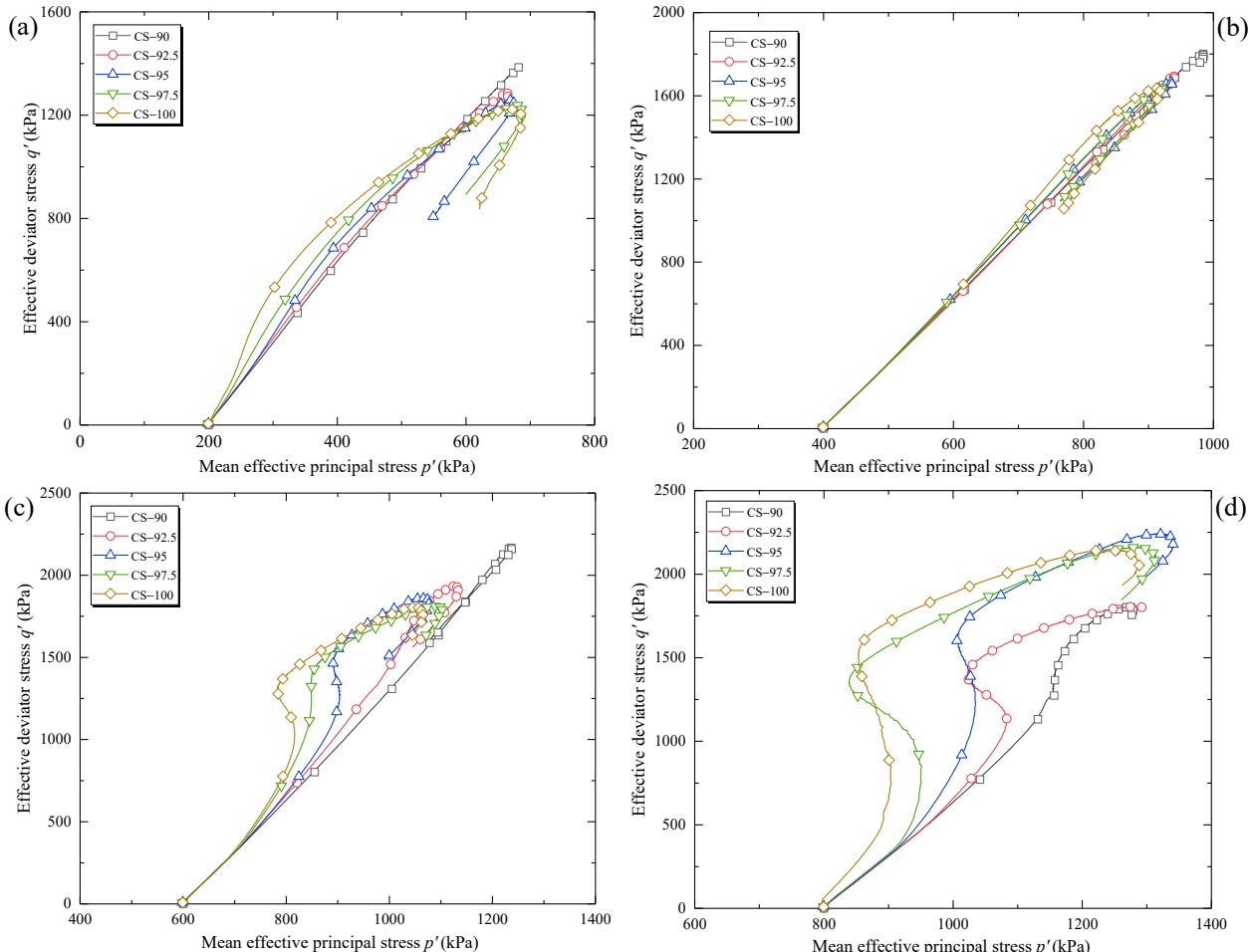

**Figure 8.** Stress path of coral sand with different saturations: (**a**) $\sigma_3' = 200$ kPa; (**b**) $\sigma_3' = 400$ kPa; (**c**) $\sigma_3' = 600$ kPa; (**d**) $\sigma_3' = 800$ kPa.

### 3.3. Initial Elastic Modulus

Initial elastic modulus ($E_i$) is an important indicator characterizing the stiffness of geomaterials, defined as the ratio of $q$ to $\varepsilon_a$ of the sample within the elastic limit [14]:

$$E_i = \frac{q}{\varepsilon_a} \tag{6}$$

Plastic deformation plays a dominant role in the deformation of coral sand under loads, while elastic deformation is minimal [42]. For this reason, the data within the range of $\varepsilon_a$ = 1% were adopted to calculate $E_i$. Figure 9 shows the $E_i$ of coral sand with $Sr$ change under different conditions. As seen from Figure 9, $E_i$ fluctuates within an extremely small range with changing $Sr$ under the same $\sigma_3'$. $E_i'$ was defined as the mean $E_i$ of coral sand samples with different $Sr$ levels under a certain $\sigma_3'$:

$$E_i' = \frac{E_{i1} + E_{i2} + \cdots + E_{in}}{n} \tag{7}$$

where $n$ is a natural constant, set at 1, 2, 3, and 4, respectively. With increasing $\sigma_3'$, $E_i'$ gradually increased from 480.1 kPa to 1046.3 kPa (Figure 9). When $\sigma_3'$ rose, the constraining force applied to the sample increased, and so did the slope of the $q$–$\varepsilon_a$ curve in the initial shear stage, as manifested by a positive correlation between $E_i'$ and $\sigma_3'$. The results in Figure 9 also indicated that, compared to $Sr$, $\sigma_3'$ exerted a more significant effect on the $E_i$ of coral sand. In UU triaxial shear stress state, $q$ grew much faster than $u$ in the initial shear stage (Figures 4 and 5). Coral sand particles bore most external loads and experienced elastic deformation. Thus, $Sr$ had no obvious effect on $E_i$.

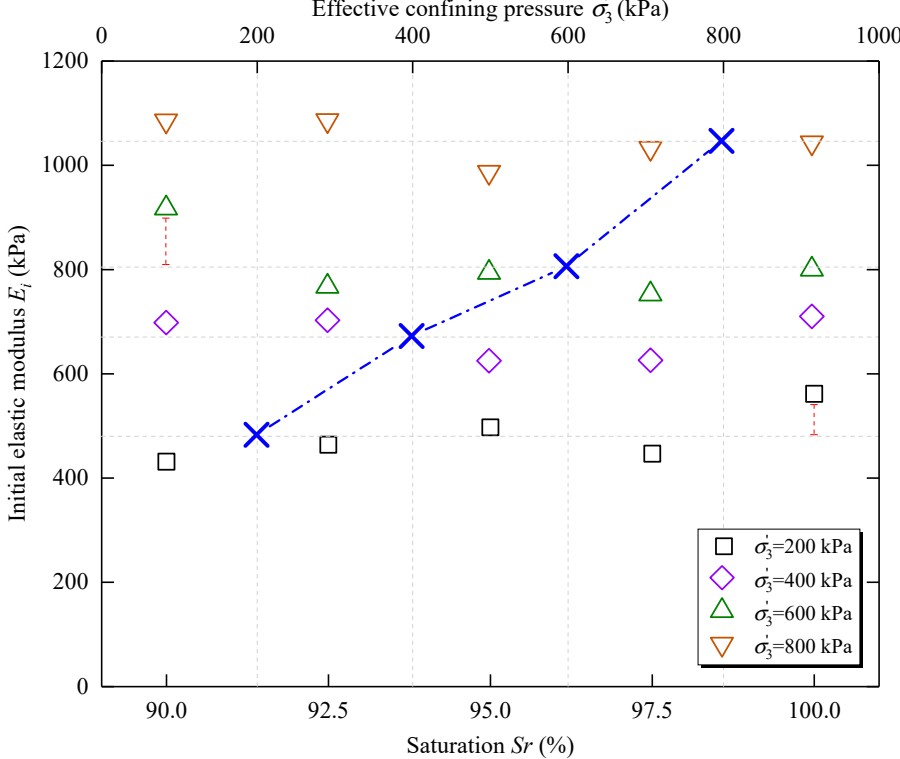

**Figure 9.** Relationship among initial elastic modulus, effective confining pressure and saturation of coral sand.

### 3.4. Shear Strength

Shear strength is an important indicator for characterizing the geomaterials' mechanical properties and a major parameter in geotechnical design [43,44]. The shear strength of a geomaterial depends on its type, meso-fabric, water content, and other factors [45–48].

Wiebe [49], Hu et al. [50], Malizia and Shakoor [51] studied terrigenous geomaterials and found that the soil shear strength decreased as water content increased. Due to its particular structure, coral sand contains numerous interparticle pores during the natural accumulation process [52], which can accommodate abundant water. Since sand particles and pore water share external loads, *Sr* will affect the shear strength of coral sand. In this section, peak deviator stress ($q_p$) was adopted to analyze the *Sr* effect of the shear strength of coral sand (Figure 10). Figure 10 indicates that the $q_p$ of coral sand decreases with increasing *Sr*. For example, when $\sigma'_3$ = 200 kPa, the $q_p$ of coral sand drops by 184.2 kPa with increasing *Sr* from 90% to 100%. When $\sigma'_3$ = 600 kPa, the $q_p$ of coral sand decreases by 363.4 kPa as the *Sr* rises from 90% to 100%. Pore water can be divided into free, bound, and capillary water, among which free water drifts through interparticle pores and has good fluidity. Nearly saturated coral sand has the highest content of free water [9]. Free water weakens the frictional resistance between coral sand particles [53], which is an important part of the strength of coral sand. The content of free water grows as *Sr* raises, for which the strength of coral sand declines with increasing *Sr*.

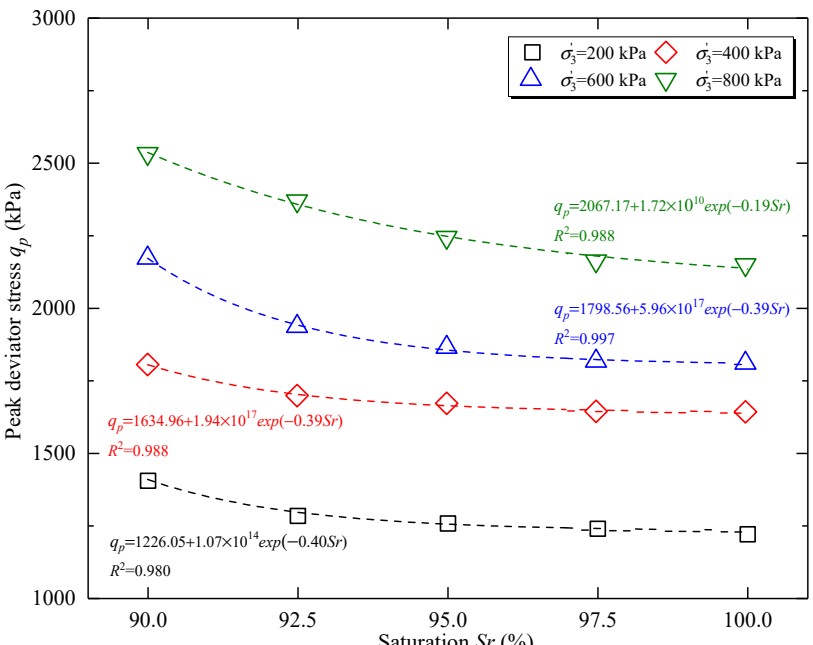

**Figure 10.** Relationship between peak deviator stress and saturation of coral sand.

Further analysis in this study found that the relationship between $q_p$ and *Sr* could be expressed as:

$$q_p = m_1 + m_2 \exp(m_3 Sr) \tag{8}$$

where $m_1$, $m_2$, and $m_3$ are fitting parameters (Table 6). $m_1$ is approximately equal to the $q_p$ of CS–100 under different $\sigma'_3$ levels. $m_1$ can characterize the shear strength of coral sand with 100% *Sr*, which is mainly affected by $\sigma'_3$. $m_2$ and $m_3$ jointly control the amplitude by which the shear strength of coral sand increases with decreasing *Sr*. $m_3$ is a negative value that characterizes the negative correlation between shear strength and *Sr*. Measurement of unsaturated shear strength at an engineering site is complex and costly. Equation (8) can be applied to predict the shear strength of unsaturated coral sand.

Figure 11 shows how the difference in $q_p$ between unsaturated coral sand and saturated coral sand changed with *Sr*, as seen in Figure 11.

**Table 6.** Peak deviator stress–saturation fitting expression and parameters.

| Sample | Expression | $\sigma_3'$ (kPa) | $m_1$ | $m_2$ | $m_3$ | $R^2$ |
|---|---|---|---|---|---|---|
| Coral medium sand | $q_p = m_1 + m_2\exp(m_3 Sr)$ | 200 | 1226.055 | $1.068 \times 10^{18}$ | $-0.404$ | 0.980 |
| | | 400 | 1634.959 | $1.939 \times 10^{17}$ | $-0.385$ | 0.988 |
| | | 600 | 1798.559 | $5.955 \times 10^{17}$ | $-0.389$ | 0.997 |
| | | 800 | 2067.165 | $1.717 \times 10^{10}$ | $-0.194$ | 0.988 |

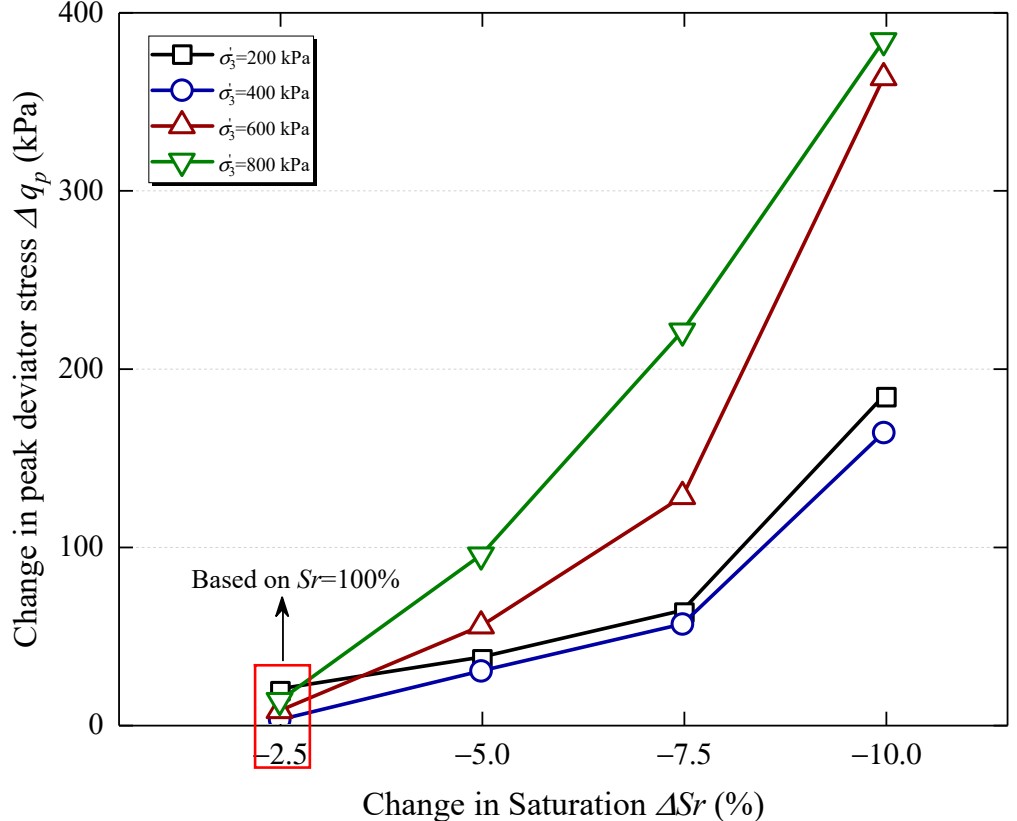

**Figure 11.** Variations in the peak deviatoric stress with saturation of coral sand.

(1) Under the same $\sigma_3'$, the difference in $q_p$ between unsaturated and saturated coral sand increased with increasing *Sr*. This law was represented in Figure 11 as the gradual increase in the slope of the linear segment, which coincided with the law expressed by Equation (8) (that is, the decreasing trend of $q_p$ with increasing *Sr* was gradually weakened). It was because increasing *Sr* weakened the shear strength of coral sand.

(2) Clearly, increasing $\sigma_3'$ grew the difference in $q_p$ between unsaturated and saturated coral sand. The *Sr* effect of the shear strength of coral sand was significantly affected by $\sigma_3'$. For example, under a low effective confining pressure ($\sigma_3' \leq 400$ kPa), when *Sr* decreased from 97.5% to 92.5%, the maximum difference in $q_p$ between unsaturated and saturated coral sand was 63.4 kPa only. In contrast, under high effective confining pressures ($\sigma_3' = 600$ and 800 kPa), when *Sr* decreased from 97.5% to 92.5%, the maximum differences in $q_p$ between unsaturated and saturated coral sand were 127.6 kPa and 220.8 kPa, respectively. Under a low effective confining pressure ($\sigma_3' \leq 400$ kPa), *u* in a coral sand sample can be disregarded in relation to its axial stress. However, increasing $\sigma_3'$ caused *u* to increase significantly, for which the ratio of *u* to axial stress also increased obviously. In other words, $\sigma_3'$ affected the shear strength of a coral sand sample by affecting *u* in the sample. For this reason, the shear strength of coral sand samples with different *Sr* levels uniformly showed an obvious confining pressure effect.

### 3.5. Shear Strength Parameter

According to the Mohr–Coulomb strength theory, the strength of a geomaterial mainly consists of two parts, frictional and cohesive strength:

$$\tau = c + \sigma \tan\varphi \tag{9}$$

where $\sigma$ is the normal stress applied on the shear plane, $\varphi$ is the internal friction angle, $c$ is cohesion. The Mohr's stress circles and shear strength envelope curves of coral sand samples with different $Sr$ levels can be depicted using Equation (9) and test results (Figure 12). As shown in Figure 12, for coral sand samples with the same $Sr$, their shear strength envelope curves are tangent to Mohr's stress circles under different $\sigma'_3$ levels. The slopes of shear strength envelope curves and the intercepts with the $\tau$ axis can be utilized to determine the shear strength parameters of coral sand samples with different $Sr$ levels.

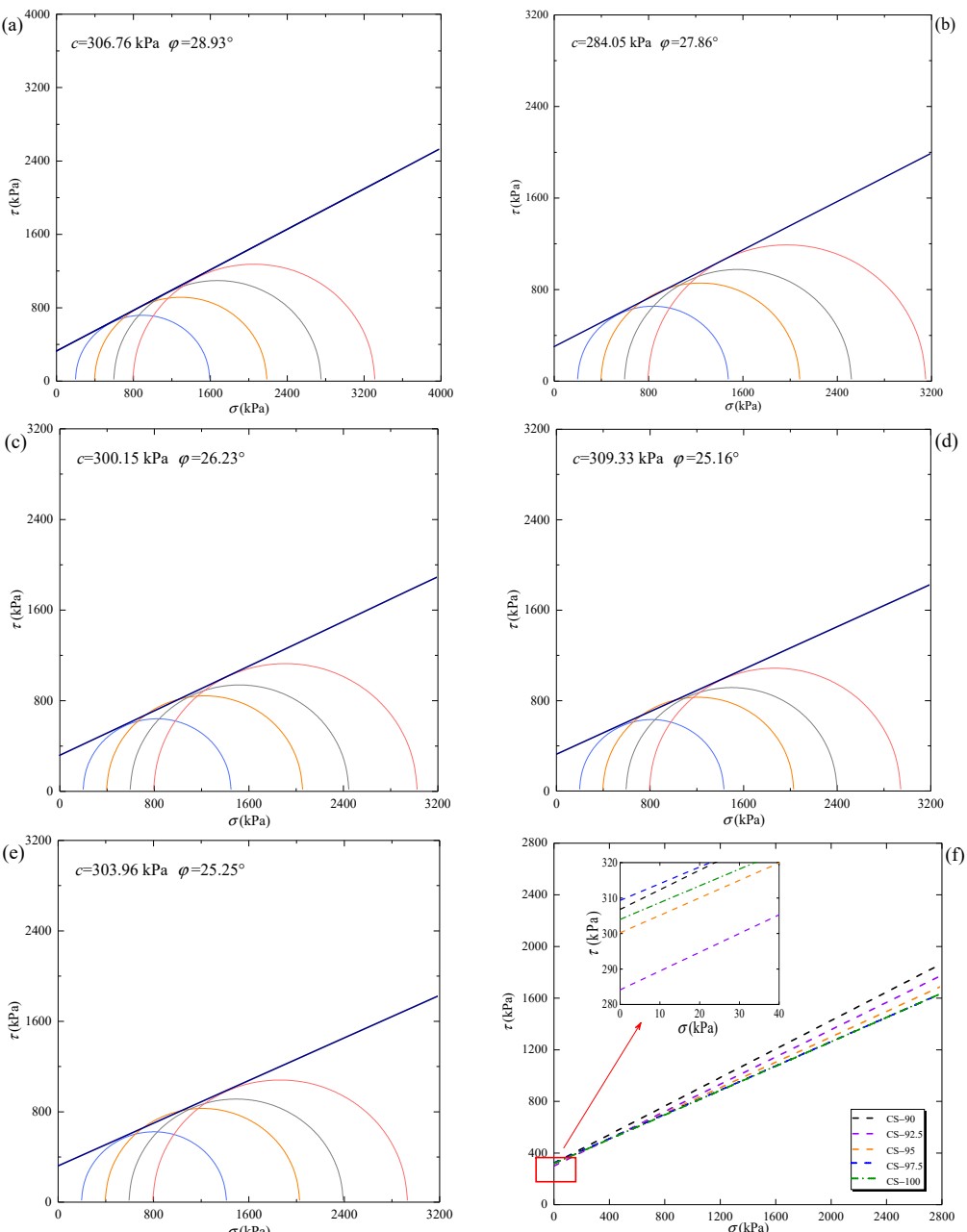

**Figure 12.** The shear strength envelope curves of: (**a**) CS–90; (**b**) CS–92.5; (**c**) CS–95; (**d**) CS–97.5; (**e**) CS–100; and (**f**) the comparison.

Figure 13 shows the relationships of the shear strength parameters of coral sand with *Sr*. According to the results in Figure 13.

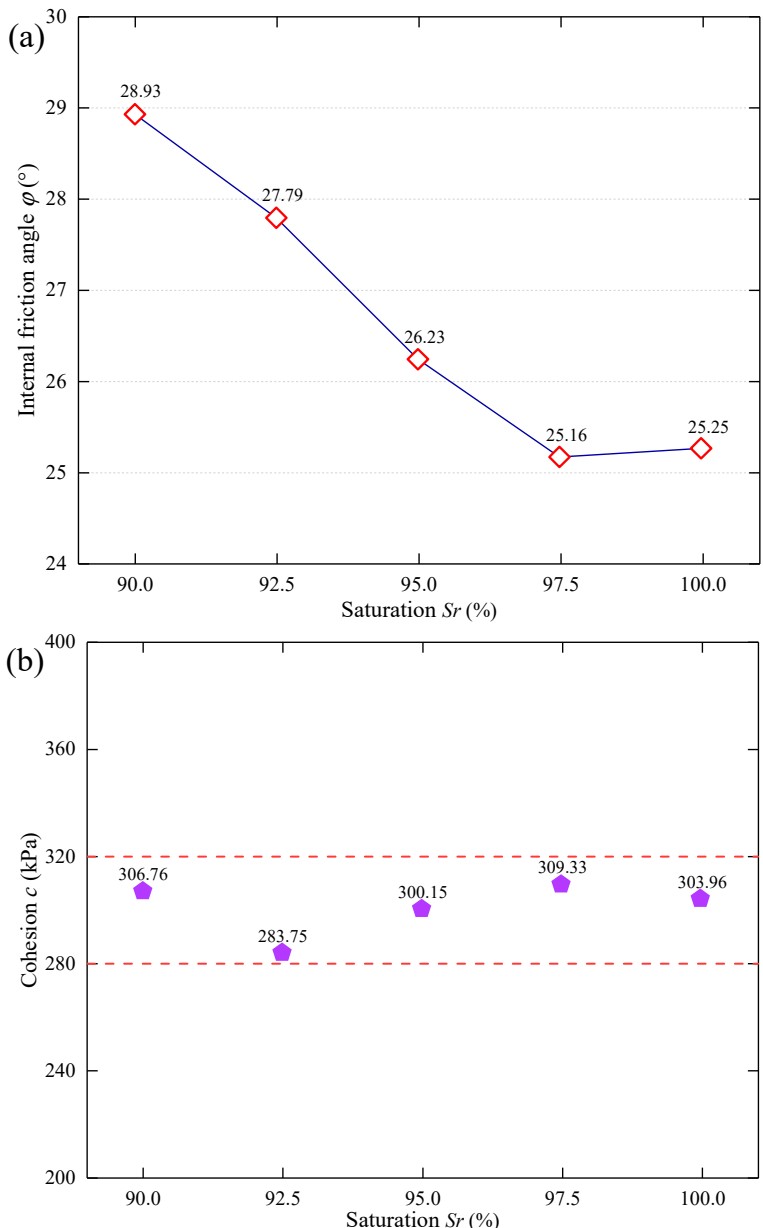

**Figure 13.** Relationship between strength parameter and saturation of coral sand: (**a**) internal friction angle; (**b**) cohesion.

(1) Within the $\sigma_3'$ range of 800 kPa, the $\varphi$ values of coral sand samples with 90–100% *Sr* in UU triaxial shear stress state fell within the range of 25–29° (Figure 13a). When *Sr* increased successively from 90% to 97.5%, the $\varphi$ of coral sand decreased approximately linearly by 3.77°. When *Sr* increased from 97.5% to 100%, $\varphi$ remained almost unchanged. Classical soil mechanics holds that the $\varphi$ of geomaterials can be expressed as [54,55]:

$$\varphi = \varphi_u + \varphi_d + \varphi_b \qquad (10)$$

where $\varphi_u$ is sliding friction angle, $\varphi_d$ is the friction angle caused by dilation, $\varphi_b$ is the friction angle caused by particle breakage and rearrangement. The lubrication effect of pore water reduces the frictional resistance between coral sand particles in relative motion, causing $\varphi_u$ to decrease as *Sr* increases. Additionally, the lubrication action of pore water intensifies

the relative motion between coral sand particles, causing particle breakage quantity to rise with increasing *Sr*. The intensification of particle breakage weakens the shear strength of coral sand [40], and $\varphi_b$, affected by particle breakage, also decreases with increasing *Sr*. In addition, under the combined action of $\sigma_3'$ and particle breakage, coral sand samples in testing exhibited a small dilation degree or not at all (Figure 5), indicating that $\varphi_d$ has a low weight in $\varphi$ and exerts a small effect. Therefore, $\varphi$ was mainly controlled by $\varphi_u$ and $\varphi_b$, which both decreased as *Sr* dropped, resulting in a negative correlation between $\varphi$ and *Sr*.

(2) Within the $\sigma_3'$ range of 800 kPa, coral sand samples with 90–100% *Sr* in UU triaxial shear stress state possessed a *c*. *c* fluctuated within a small range with changing *Sr*, but no apparent law of change was observed. *c* had a range of 280–310 kPa and was mainly concentrated between 300 and 310 kPa. The maximum difference in *c* between coral sand samples with various *Sr* levels was 25.3 kPa only (Figure 13b). According to classical soil mechanics, sandy soil has no cohesion. However, there is capillarity attraction in unsaturated sandy soil, and suitable water content produces an obvious cohesive action. The cohesion existing in these forms is also called "pseudo cohesion" or "apparent cohesion" [56]. The change in *Sr* affects the content and distribution of capillary water in coral sand, so *Sr* affects the apparent cohesion of coral sand. Meanwhile, in the $\tau$–$\sigma$ stress space, the UU shear strength envelope curves of coral sand samples with different *Sr* levels were almost parallel to the $\sigma$ axis, so the *c* corresponding to the intercept of coral sand changed a little.

## 4. Particle Breakage Analysis

### 4.1. Effect of Saturation on Particle Breakage Ratio

The particle gradation curves of geomaterials change upon breakage. The particle breakage degree of a geomaterial can be quantified by the differences in its particle gradation curves before and after testing [57]. Hardin [58] mentioned that geomaterials with different initial gradations break into powder particles with sizes of less than 0.074 mm under sufficiently large external loads. On this basis, Hardin proposed a particle breakage evaluation indicator, i.e., relative breakage ratio ($B_r$):

$$B_r = \frac{B_t}{B_p} \tag{11}$$

where $B_t$ denotes the total breakage, which is the area enclosed by the particle gradation curves before and after testing and the vertical line of $d = 0.074$ mm, and $B_p$ denotes breakage potential, which is the area enclosed by the particle gradation curve before testing and the vertical line of $d = 0.074$ mm (Figure 14).

The $B_r$ of coral sand under each condition was calculated using Equation (11) and the results of particle screening tests. Figure 15 shows the relationship between $B_r$ and *Sr* for coral sand in the UU triaxial shear stress state. As can be seen from Figure 15, $B_r$ increase with increasing *Sr*, which coincides with the research results of Wu et al. [9]. A coral sand sample with a higher *Sr* level has a higher pore water content, producing a lubrication action in interparticle relative motion. The lubrication action of pore water makes it easier for coral sand particles to experience rolling and displacement. In contrast, the increase in the probability of coral sand particles results in fracturing, abrading, or otherwise breaking them into relative motion, which causes a higher $B_r$.

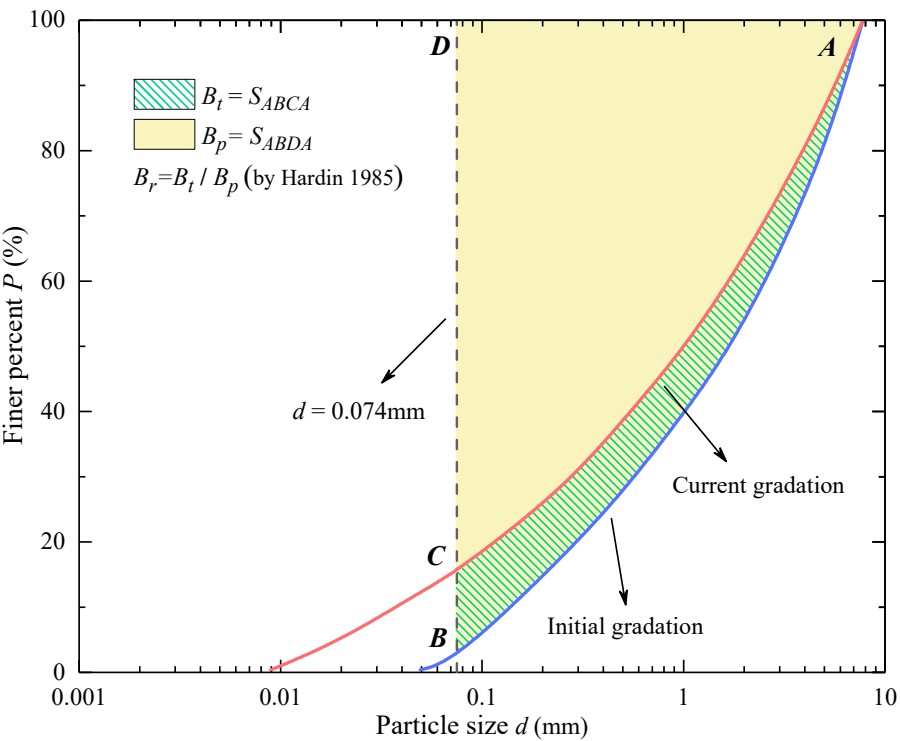

**Figure 14.** Calculated method of relative breakage ratio $B_r$.

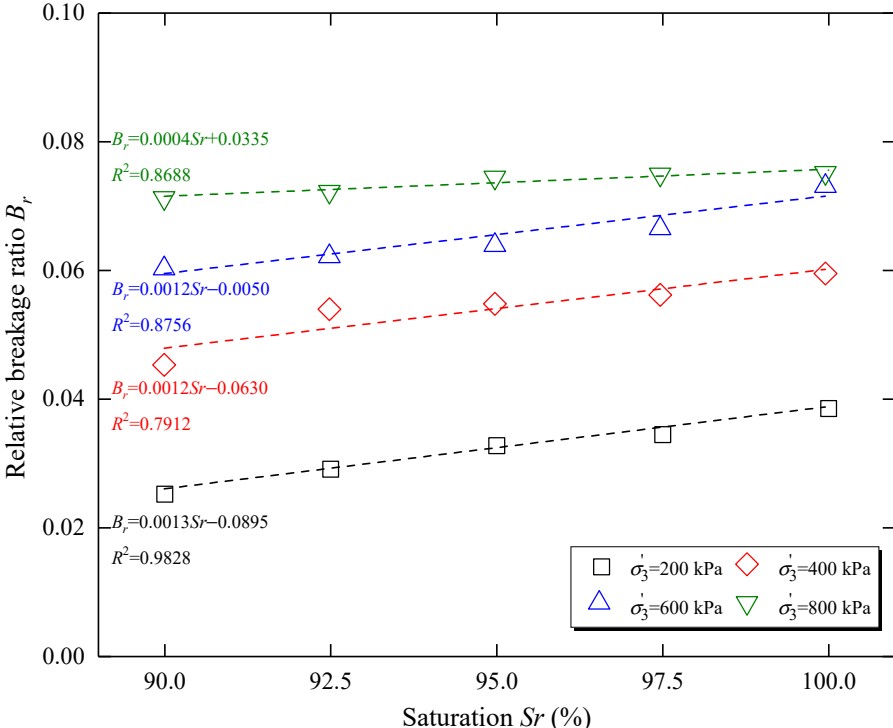

**Figure 15.** Relationship between relative breakage ratio and saturation of coral sand.

Further analysis revealed a good linear relationship between $B_r$ and $Sr$ for coral sand, which could be expressed as:

$$B_r = p_1 Sr + p_2 \tag{12}$$

where $p_1$ and $p_2$ are fitting parameters, $p_1$ is a positive value that characterizes the degree to which the $B_r$ of coral sand increases with increasing $Sr$, and $p_2$ characterizes the combined

effect of other factors (such as compactness, stress level, and particle gradation) on the $B_r$ of coral sand. The $Sr$ effects on coral sand particle breakage disappeared when $\sigma'_3$ was sufficiently large. For example, when $\sigma'_3$ = 200, 400, and 600 kPa, $p_1$ was 0.0012, 0.0012, and 0.0013, respectively, and the fitting curves were parallel to each other. When $\sigma'_3$ = 800 kPa, $p_1$ was 0.0004, and $B_r$ increased very slightly with increasing $Sr$. When $\sigma'_3$ was sufficiently large, the intensification of particle breakage produced more fine particles that can be used to fill interparticle spacings. Thus, the presence of these fine particles obstructed the flow of pore water and further weakened its lubrication action on interparticle relative movement. This explains why $Sr$ exerts little effect on the particle breakage of coral sand under a high value of $\sigma'_3$. In addition, the production of fine particles caused particle breakage to stabilize by increasing $\sigma'_3$ gradually, which means that the particle breakage cannot proceed infinitely with increasing $\sigma'_3$ [59]. This phenomenon is pronounced at high $Sr$ coral sand (Figure 15).

In the studies of coral sand particle breakage, saturated coral sand was the main target. In view of this, particle breakage results of coral sand with 100% $Sr$ in this test are selected to compare with those introduced from other tests [18,25,27–29,33,60]. Figure 16 shows the particle breakage results of saturated coral sand under different conditions. It can be seen from Figure 16 that, $B_r$ of coral sand is different due to the influence of sand origin, particle gradation, drained condition, compactness, and so on. Within the $\sigma'_3$ range of 800 kPa, $B_r$ of coral sand is mainly distributed between 0.01 and 0.12, and $B_r$ shows a significant positive correlation with $\sigma'_3$. On the whole, $B_r$ of coral sand in this test is relatively small, mainly for the following reasons: (I) The compactness of sand sample undergoes isotropic compression increases, and $B_r$ increases accordingly. While sand sample in this test was not consolidated, resulting in a decrease of $B_r$. (II) Figure 16 illustrates that $B_r$ of coral sand under drained condition is greater than under undrained condition, and this test was conducted under undrained condition, so $B_r$ is relatively small. (III) Coral sand with finer particle size has less potential to breakage [17,18]. Coral sand used in this test was medium size, which is finer than sand in other tests, so $B_r$ in the test is smaller.

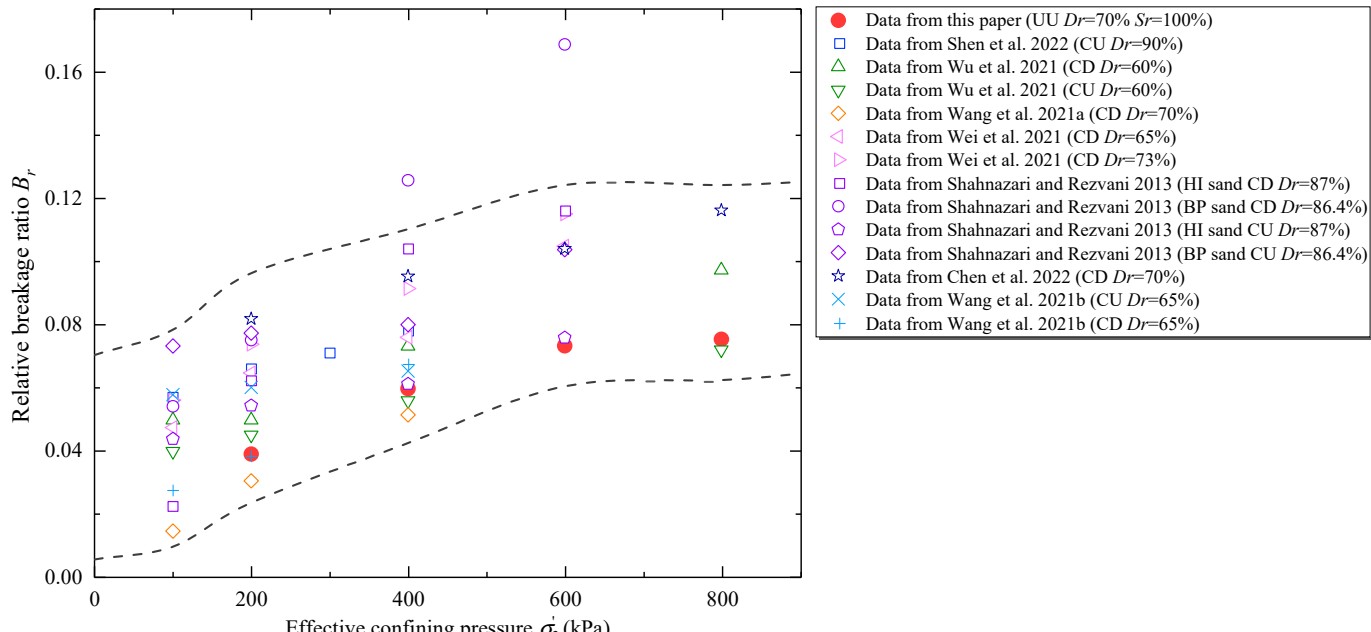

**Figure 16.** Previous test results of particle breakage of coral sand [25,27,28,33,39,60].

Furthermore, Wu et al. [9] measured the particle breakage of coral sand with different moisture contents based on CD triaxial shear test. Figure 17 shows the particle breakage results of Wu and this paper. The test result in this paper illustrated that there was a positive relationship between $B_r$ and $Sr$ for coral sand with $Sr \geq 90\%$. Test results in Wu' paper showed that there was also a positive relationship between $B_r$ and $Sr$ for coral sand

with *Sr* ranging from 17 to 69%. Further analysis indicates that Wu' data and the data in this paper can be described by the same linear expression with a great fit. In above, in can be inferred that there is a linear relationship between $B_r$ and *Sr* of coral sand, and the relationship does not rely on *Sr* level of coral sand. However, the conditions of Wu' test are different from those in this paper (such as particle gradation, drained condition), and this speculation still needs to be verified by further experiments.

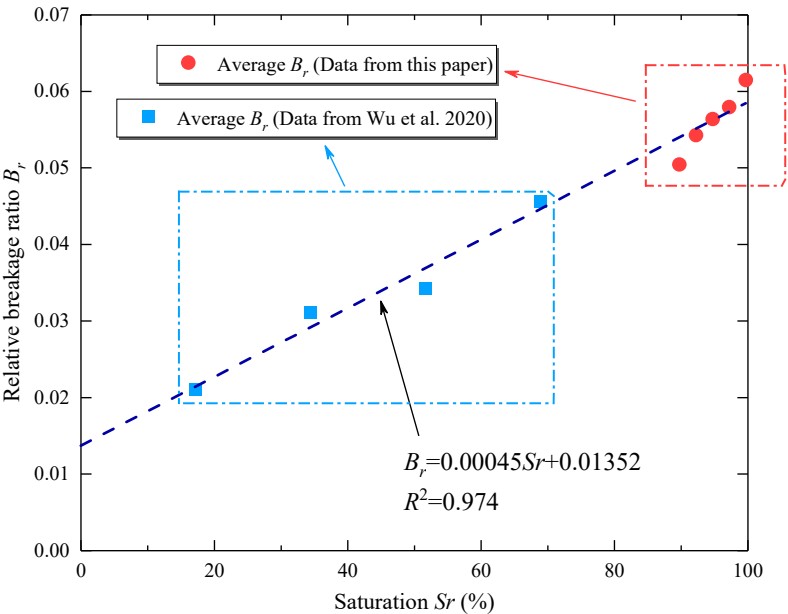

**Figure 17.** Relationship between relative breakage ratio and saturation combined with data from Wu et al. [9].

### 4.2. Effect of Saturation on Modified Particle Breakage Index

The calculation of $B_r$ is based on the condition that the particle breakage limit is *d* = 0.074 mm. However, this condition is controversial. Subsequent studies showed that particles do not completely break into powder particles even when subjected to high stresses with values of about 689 MPa [61]. Einav [62] claimed that the limit of particle breakage obeys the following function:

$$f(d) = \left(\frac{d}{d_M}\right)^{3-\alpha} \tag{13}$$

where $f(d)$ denotes the ultimate particle gradation curve, $d_M$ denotes the upper limit of particle size, and $\alpha$ denotes fractal dimension, set at 2.6 [63]. On this basis, Einav introduced the modified relative breakage index ($B_r^*$):

$$B_r^* = \frac{B_t}{B_p^*} \tag{14}$$

where $B_t$ denotes total breakage, the value of which is the area enclosed by the initial particle gradation curve and the current particle gradation curve, and $B_p^*$ denotes modified breakage potential, the value of which is the area enclosed by the initial particle gradation curve and the ultimate particle gradation curve (Figure 18).

The coral sand $B_r^*$ under each condition was calculated according to Equation (14) and the results of particle screening tests. Figure 19 shows the relationship between $B_r^*$ and *Sr* for coral sand in UU triaxial shear stress state. Under the same value of $\sigma_3'$, the $B_r^*$ of coral sand increased linearly with increasing *Sr*. When $\sigma_3'$ was sufficiently large, the growth rate of $B_r^*$ with increasing *Sr* dropped significantly. As particle breakage evaluation indicators, $B_r$ and $B_r^*$ were similar in some aspects but different in others [29]. $B_r$ and $B_r^*$

were similar in that they can describe the objective laws governing the particle breakage of coral sand in the same manner. $B_r$ and $B_r^*$ both depend on the initial particle gradation curve and the current particle gradation curve for calculating the actual particle breakage quantity. The initial particle gradation curve and the current particle gradation curve were definite values under given test conditions. They were different in that $B_r^*$ was always greater than $B_r$ under the same test conditions. A comparison between Equations (11) and (14) revealed that their main difference was the denominators. Due to the difference in ultimate gradation, $B_p^*$ was always smaller than $B_p$ under the same conditions, which was the main reason why $B_r^*$ was greater.

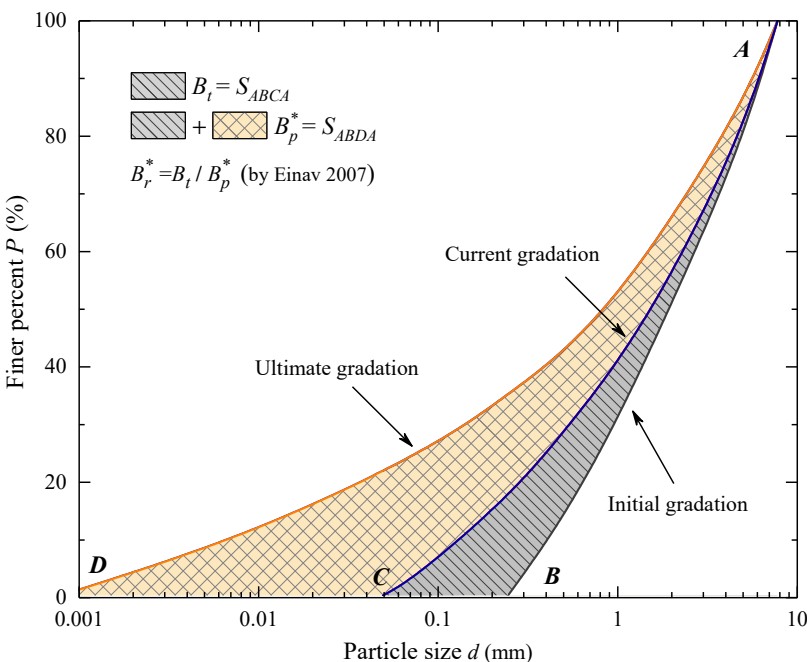

**Figure 18.** Calculated method of modified relative breakage index $B_r^*$.

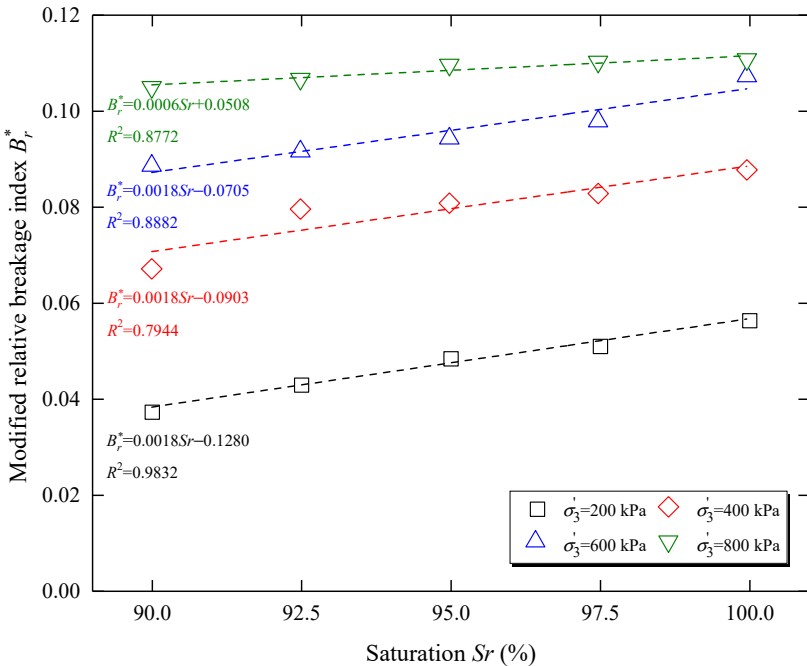

**Figure 19.** Relationship between modified relative breakage index and saturation of coral sand.

### 4.3. Effect of Saturation on Particle Median Size

Coral sand samples with different *Sr* levels experienced varying degrees of particle breakage in the shear process. Since some sand particles would break into finer sand particles, the median particle size ($d_{50}$) of samples would change accordingly. After testing, the $d_{50}$ values of coral sand samples were calculated and then statistically investigated. Additionally, box plots were used to present the law of $d_{50}$ changing with *Sr*. As seen from Figure 20, the mean and median $d_{50}$ values of coral sand both decrease with increasing *Sr*. That is, under the same conditions, a sample with a higher *Sr* level can produce more fine particles after shear tests, suggesting that a higher *Sr* level means a larger particle breakage quantity for coral sand. This result is consistent with the result in Figure 15 (or Figure 19).

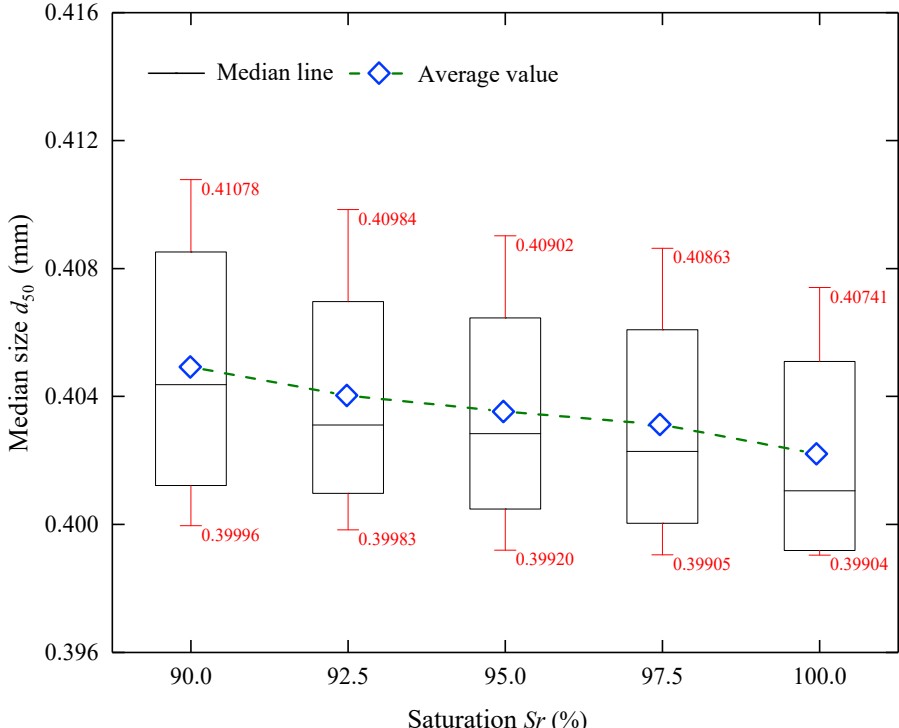

**Figure 20.** Relationship between median particles size and saturation of coral sand.

Figure 21 shows the relationships of $d_{50}$ with $B_r$ and $B_r^*$. It can be seen from Figure 21 that $d_{50}$ decreases with increasing $B_r$ and $B_r^*$, with linear fitting degrees of 0.9994 and 0.9989, respectively. A sample with a large particle breakage degree would produce more particles in the shear process, and consequently, the particle size corresponding to the cumulative percentage of particle size contents of 50% would be smaller. Thus, it can be observed that there is a negative correlation between $d_{50}$ and particle breakage quantity.

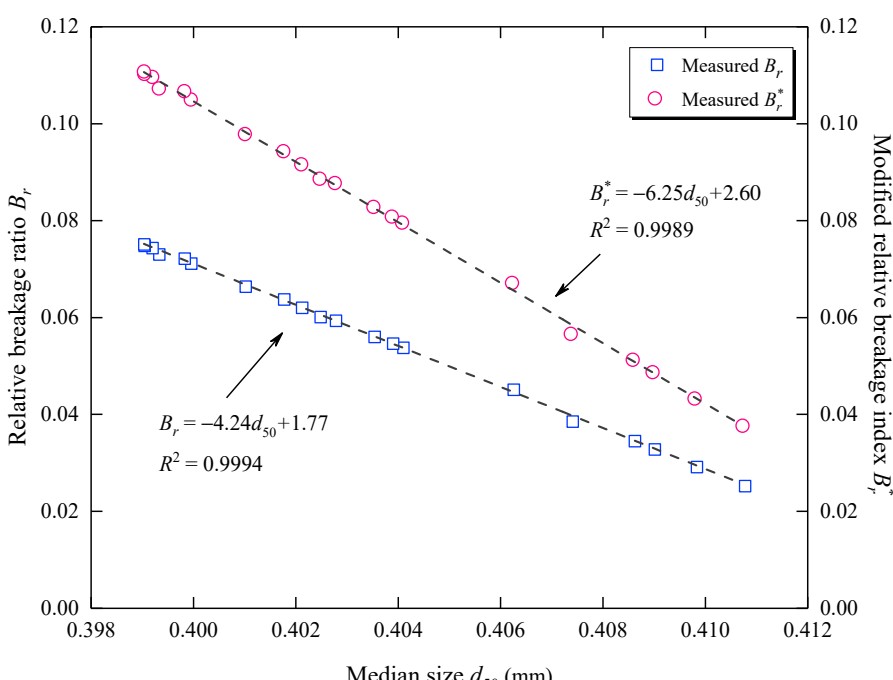

**Figure 21.** Relationship between relative breakage ratio, modified relative breakage index, and median particle size of coral sand.

## 5. Discussion of Saturation Effect on Shear Strength and Particle Breakage

### 5.1. Saturation Effect on Shear Strength of Coral Sand

The strength characteristics of coral sand samples with different $Sr$ levels (90, 92.5, 95, 97.5, and 100%) are discussed. It was found that the strength characteristics of coral sand are basically consistent under 97.5% $Sr$ and 100% $Sr$ (Table 7). Therefore, 97.5% can be taken as the critical $Sr$ of coral sand from the strength perspective. The mechanical properties of coral sand become stable after reaching the critical $Sr$, in that case, further increasing $Sr$ produces no practical significance. In contrast, the $Sr$ effect of the mechanical properties of coral sand is significantly below the critical $Sr$.

**Table 7.** Comparison in shear behavior between coral sand with 97.5% saturation and 100% saturation.

| Shear Behavior | | Saturation | |
|---|---|---|---|
| | | $Sr = 97.5\%$ | $Sr = 100\%$ |
| Shear strength $q_p$ (kPa) | $\sigma_3' = 200$ kPa | 1240.1 | 1220.9 |
| | $\sigma_3' = 400$ kPa | 1639.9 | 1638.1 |
| | $\sigma_3' = 600$ kPa | 1813.2 | 1806.2 |
| | $\sigma_3' = 800$ kPa | 2159.7 | 2147.1 |
| Internal fiction angle $\varphi$ (°) | | 25.16 | 25.25 |
| cohesion $c$ (kPa) | | 309.33 | 303.96 |

In view of this, Equation (15) was used to calculate the ratio of the difference between the shear strength of coral sand below the critical $Sr$ and that of coral sand at the critical $Sr$ as follows:

$$\eta = \frac{(q_p)_i - (q_p)_{97.5}}{(q_p)_{97.5}} \times 100\% \qquad (15)$$

where $\eta$ denotes the ratio at which the shear strength of coral sand increases with decreasing $Sr$, and $(q_p)_i$ denotes the $q_p$ of coral sand with a specific $Sr$ (where $i$ was set at 90, 92.5, and 95, respectively). Figure 22 shows the relationship between $\eta$ and $Sr$. $\sigma_3'$ was taken as the threshold to calculate mean $\eta$ and plot it in the figure. As seen in Figure 22, the $\eta$ values for

coral sand with 90% *Sr* fall within the range of 10–20%. At this point, the shear strength of the sample, compared with that of coral sand with the critical *Sr*, is greatly improved, and can easily cause stress that exceeds the sensor range in the test process. Additionally, it can be seen that the $\eta$ values of coral sand with 95% *Sr* are all less than 5%, suggesting that the shear strength of the sample with 95% *Sr* was extremely close to that with the critical *Sr*. The mean $\eta$ value of coral sand with 92.5% *Sr* was 3.43% under a low effective confining pressure ($\sigma'_3 \leq 400$ kPa), and 8.14% under high effective confining pressures ($\sigma'_3 = 600$ and 800 kPa).

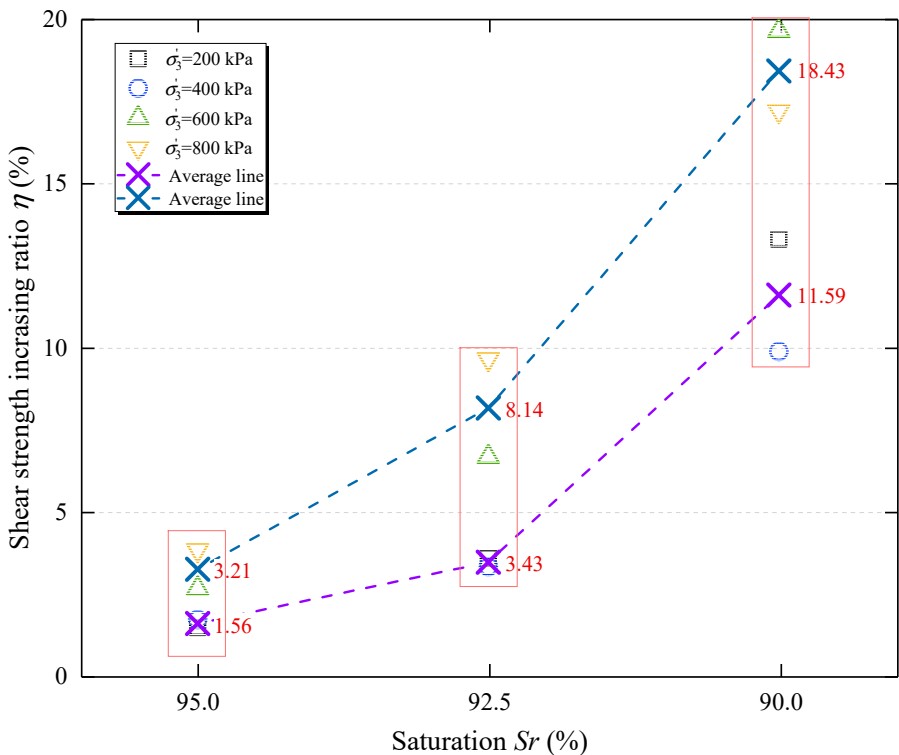

**Figure 22.** Relationship between shear strength increasing ratio and saturation of coral sand.

The preparation of saturated coral sand usually requires back pressure saturation to increase *Sr*, and *B* > 0.95 is often used as the criterion for the saturation of coral sand. To that end, it is often necessary to apply very high back pressure on coral sand [14,25,28,33,59,64]. However, coral sand is a special geomaterial that is susceptible to particle breakage even under normal stress. Accordingly, excessively high back pressure greatly modifies the original gradation of coral sand, causing changes in the mechanical properties of coral sand [65,66]. This study discusses the *Sr* effect on the strength of coral sand and aims to find a suitable *Sr* in an unsaturated region. It was found that the strength characteristics of coral sand with this *Sr* are roughly similar to those in the saturated state and that the applied back pressure on the sample at this point declines considerably (the *B* value was even smaller). On the basis of the above analysis, 92.5% *Sr* can be considered a suitable *Sr* value. Under this *Sr*, the strength of coral sand is close to that of saturated coral sand, and the *B* value corresponding to this *Sr* is 0.27, which is far less than the 0.95 required in the test code. Taking 92.5% as the *Sr* value of coral sand available for tests not only avoids the back pressure effect in the saturation process but also offers an idea for seeking a suitable saturation method for coral sand.

### 5.2. Saturation Effect on Particle Breakage of Coral Sand

Section 5.1 proposes a *Sr* value of coral sand available for tests. This section further demonstrates this conclusion from the perspective of particle breakage. Particle breakage is an irreversible physical process of particles splitting into fine particles, affecting geomateri-

als' strength, deformation characteristics, and energy absorption [67–70]. According to the calculation results in Section 4, when $\sigma_3' = 200, 400, 600$, and 800 kPa, the particle breakage quantities of coral sand with 92.5% $Sr$ in the shear process are 0.00538, 0.00224, 0.00439, and 0.00268 less than that of the coral sand with the critical $Sr$, respectively. The particle breakage quantity of coral sand was smaller under a low $Sr$, in which case particle breakage exerted an even smaller effect on the strength characteristics of coral sand. Moreover, taking 92.5% as the $Sr$ value of the coral sand available for tests in the preparation of coral sand can prevent particle breakage caused by the application of back pressure. The above discussion verifies the reasonableness of taking 92.5% as the $Sr$ value of coral sand available for tests from the perspective of particle breakage.

## 6. Conclusions

To ascertain the saturation effect of the shear behavior of coral sand, this study conducted multiple groups of UU triaxial shear tests on coral sand samples with different $Sr$ levels, and tested the particle breakage quantities of coral sand. The main conclusions of this study are as follows:

(1) Coral sand samples with different $Sr$ levels all manifested strain softening characteristics. Increasing $Sr$ promoted the development of $u_{p1}$ in coral sand during the initial shear stage but had no effects on the $E_i$ of coral sand.

(2) The shear strength of coral sand decreased exponentially with increasing $Sr$. The $\varphi$ of coral sand decreased by 3.77° with the increase of $Sr$ from 90% to 97.5%. However, when $Sr$ increased from 97.5% to 100%, the $\varphi$ of coral sand remained unchanged. The $c$ of coral sand in this study fluctuated within the range of 280–310 kPa but was not considerably affected by $Sr$.

(3) For $\sigma_3'$ range of 600 kPa, the particle breakage quantity of coral sand increased linearly at a slope of 0.12% with increasing $Sr$. As the $\sigma_3'$ continued to rise, the linear increase rate of particle breakage quantity with increasing $Sr$ declined to 0.04%. After testing, the $d_{50}$ of coral sand decreased with increasing $Sr$ or particle breakage quantity.

(4) 97.5% could be taken as the critical $Sr$ of coral sand. Above this critical $Sr$, the mechanical properties of coral sand were stable. In contrast, below this critical $Sr$, the mechanical properties of coral sand were significantly affected by $Sr$. On the basis of comparing the strength and particle breakage characteristics of unsaturated coral sand and coral sand in a critical saturation state, this study suggested that 92.5% should be taken as the $Sr$ value of coral sand available for tests.

**Author Contributions:** X.C.: Conceptualization, methodology, and writing—original draft; J.S.: investigation and project administration; X.W.: formal analysis and writing—original draft; T.Y.: writing—review and editing; D.X.: writing—review and editing. All authors have read and agreed to the published version of the manuscript.

**Funding:** This research was funded by the National Natural Science Foundation of China (Nos. 41772336, 42177151, and 42177154).

**Data Availability Statement:** Some or all data, models, or codes generated or used during this study are available from the corresponding author upon request.

**Conflicts of Interest:** The authors declare on conflict of interest.

## Nomenclature

| | |
|---|---|
| CS | Coral sand |
| $Sr$ | Saturation (Unit: %) |
| $\sigma_3'$ | Effective confining pressure (Unit: kPa) |
| UU | Unconsolidated–undrained |
| CD | Consolidated–drained |
| CU | Consolidated-undrained |
| $d$ | Particle size (Unit: mm) |
| $d_{50}$ | Diameter that corresponds to an 50% finer in the particle gradation curve (Unit: mm) |
| $\rho_d$ | Dry density (Unit: g/cm$^3$) |
| $e$ | Void ratio |
| $P$ | Mass percentage of particles smaller than a certain size (Unit: %) |
| $Cu$ | Nonuniformity coefficient |
| $Cc$ | Curvature coefficient |
| $Gs$ | Specific gravity |
| $\rho_{dmin}$ | Minimum dry density (Unit: g/cm$^3$) |
| $\rho_{dmax}$ | Maximum dry density (Unit: g/cm$^3$) |
| $e_{min}$ | Minimum void ratio |
| $e_{max}$ | Maximum void ratio |
| $Dr$ | Relative density (Unit: %) |
| $v$ | Shear rate (Unit: mm/min) |
| $\varepsilon_a$ | Axial strain (Unit: %) |
| $n$ | Porosity |
| $V_W$ | Volume of water inside sample (Unit: mm$^3$) |
| $V_V$ | Volume of pores inside sample (Unit: mm$^3$) |
| $V$ | Sample volume (Unit: mm$^3$) |
| $q$ | Deviator stress (Unit: kPa) |
| $q_p$ | Peak deviator stress (Unit: kPa) |
| $u$ | Pore water stress (Unit: kPa) |
| $u_{p1}$ | Peak pore water stress (Unit: kPa) |
| $\sigma_1'$ | Effective major principal stress (Unit: kPa) |
| $p'$ | Mean effective principal stress (Unit: kPa) |
| $E_i$ | Initial elastic modulus (Unit: kPa) |
| $E_i'$ | Mean initial elastic modulus (Unit: kPa) |
| $m_1, m_2, m_3$ | Fitting parameters |
| $\tau$ | Shear stress acting on the shear plane (Unit: kPa) |
| $\sigma$ | Normal stress on the shear plane (Unit: kPa) |
| $\varphi$ | Internal friction angle (Unit: °) |
| $c$ | Cohesion (Unit: kPa) |
| $\varphi_u$ | Sliding friction angle (Unit: °) |
| $\varphi_d$ | Dilatancy friction angle (Unit: °) |
| $\varphi_b$ | Friction angle caused by particle breakage and rearrangement (Unit: °) |
| $B_r$ | Relative breakage ratio |
| $B_t$ | Total breakage |
| $B_p$ | Breakage potential |
| $p_1, p_2$ | Fitting parameters |
| $\alpha$ | Fractal dimension |
| $d_M$ | Upper limit of particle size (Unit: mm) |
| $B_p^*$ | Modified breakage potential |
| $B_r^*$ | Modified relative breakage index |
| $\eta$ | Shear strength increasing ratio (Unit: %) |
| $B$ | Pore water stress index |

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
