# Peer review of "Effect of Saturation on Shear Behavior and Particle Breakage of Coral Sand"

_jmse, doi:10.3390/jmse10091280_

Round 1

Reviewer 1 Report

In the paper, the effect of saturation on shear behavior and particle breakage of coral sand is experimentally examined. Authors extensively reviewed the available literature related to the this study.

In order to draw reasonable conclusions, the author needs to compare and analyze not only the experimental results but also the research results of existing researchers.

In order for this paper to be published, comparative analysis between the experimental results from the literature analyzed in the introduction  and this test results conducted by authors should be conducted and the rational  conclusion should be drawn based on the comparison.

Reviewer 2 Report

The paper is well writer and with an in depth analysis of the methodological aspects of the research and of the obtained results.

The only aspects that must be improved are the mistakes in the introduction presentation starting from line 49 and the figures that must be necessarily presented with error bars.

For this reason, I suggest to accept this paper with minor revisions.

Reviewer 3 Report

In the presented article, the strength parameters of sandy soil are determined depending on the degree of water saturation of more than 90 percent. It is shown that this factor practically does not affect the deformation modulus and slightly affects the angle of internal friction. Linear and exponential dependences of strength on the degree of water saturation and hydrostatic pressure are shown.

As a result, the article may be useful for a more complete understanding of the operation of this type of squeak at high degrees of water saturation, including for further mathematical description. It is worth noting that the results obtained are obvious and have value only in working with the specific type of sand presented.
